# Why plants make puzzle cells, and how their shape emerges

Aleksandra Sapala[1†], Adam Runions[1,2†], Anne-Lise Routier-Kierzkowska[1‡], Mainak Das Gupta[1,3], Lilan Hong[4,5], Hugo Hofhuis[1], Stéphane Verger[6], Gabriella Mosca[1§], Chun-Biu Li[7], Angela Hay[1], Olivier Hamant[6], Adrienne HK Roeder[4,5], Miltos Tsiantis[1], Przemyslaw Prusinkiewicz[2], Richard S Smith[1]*

[1]Department of Comparative Development and Genetics, Max Planck Institute for Plant Breeding Research, Cologne, Germany; [2]Department of Computer Science, University of Calgary, Calgary, Canada; [3]Department of Microbiology and Cell Biology, Indian Institute of Science, Bangalore, India; [4]Weill Institute for Cell and Molecular Biology, Cornell University, Ithaca, United States; [5]School of Integrative Plant Science, Section of Plant Biology, Cornell University, Ithaca, United States; [6]Laboratoire Reproduction et Développement des Plantes, Université de Lyon, ENS de Lyon, UCBL, INRA, CNRS, Lyon, France; [7]Department of Mathematics, Stockholm University, Stockholm, Sweden

*For correspondence:
smith@mpipz.mpg.de

[†]These authors contributed equally to this work

Present address: [‡]Institut de Recherche en Biologie Végétale, Université de Montréal, Montréal, Canada; [§]Department of Plant and Microbial Biology, University of Zürich, Zürich, Switzerland

Competing interests: The authors declare that no competing interests exist.

**Abstract** The shape and function of plant cells are often highly interdependent. The puzzle-shaped cells that appear in the epidermis of many plants are a striking example of a complex cell shape, however their functional benefit has remained elusive. We propose that these intricate forms provide an effective strategy to reduce mechanical stress in the cell wall of the epidermis. When tissue-level growth is isotropic, we hypothesize that lobes emerge at the cellular level to prevent formation of large isodiametric cells that would bulge under the stress produced by turgor pressure. Data from various plant organs and species support the relationship between lobes and growth isotropy, which we test with mutants where growth direction is perturbed. Using simulation models we show that a mechanism actively regulating cellular stress plausibly reproduces the development of epidermal cell shape. Together, our results suggest that mechanical stress is a key driver of cell-shape morphogenesis.

DOI: https://doi.org/10.7554/eLife.32794.001

## Introduction

During growth and morphogenesis, plant cells undergo dramatic changes in size and shape. Starting from small isodiametric cells in proliferative tissues, cells stop dividing and can expand to over 100 times their original size. This results in large elongated cells, such as those in roots and stems, or much more intricate forms, such as the jigsaw puzzle-shaped epidermal cells of *Arabidopsis thaliana* leaves and cotyledon (*Figure 1A*), which we call puzzle cells. The processes underlying the formation of these cells are presently unclear, and it has been proposed that they emerge from either the localized outgrowth of lobes (also called protrusions) (*Fu, 2002*; *Mathur, 2006*; *Xu et al., 2010*; *Zhang et al., 2011*), localized restriction of indentations (*Fu et al., 2009*; *Sampathkumar et al., 2014*; *Lin et al., 2013*), or a combination of both (*Fu et al., 2005*; *Abley et al., 2013*; *Armour et al., 2015*; *Higaki et al., 2016*; *Majda et al., 2017*). Specific members of the Rho GTPase of plants (ROP) family of proteins play a key role in shaping these cells. ROP2 and ROP6 mutually inhibit each other's accumulation at the plasma membrane, creating a co-repression network that

**eLife digest** Cells with complex interlocking shapes, similar to pieces of a jigsaw puzzle, cover the surface of many leaves. Why do these curious shapes form, and what benefit do they provide to the plant?

Plant cells are like small balloons surrounded by a strong cell wall. Their internal pressure can be higher than the pressure in a car tire. It is this pressure that gives non-woody plant tissue its shape. Take away the pressure, and the plant wilts.

The pressure inside a cell creates a lot of mechanical stress on the epidermal cell walls – those that make up the surface of the plant. The extent of the stress depends on the shape and size of the cells; for example, large cells bulge out and experience more stress than small cells. This could mean that the shape of puzzle cells is an adaptation used by plants to reduce the stress on their surface.

To investigate this possibility, Sapala, Runions et al. developed a computer simulation that models how a plant grows and re-creates a variety of realistic puzzle cell shapes. The simulations show that 'paving' the leaf surface with puzzle shaped cells instead of more regularly shaped cells reduces the stress in the epidermal cell walls.

Counterintuitively, the simulations also show that complex puzzle shapes develop in parts of the plant that grow isotropically (uniformly in all directions), such as leaves. If a plant organ grows mostly in one direction, like in a root or stem, long thin cells are sufficient to reduce the stress on the epidermal cell wall. Sapala, Runions et al. tested this idea by analyzing the shape of organs and cells in many plant species and by genetically modifying growth directions in *Arabidopsis thaliana* plants. This confirmed that puzzle cell shape is related to both organ shape and how isotropically the plant grows.

It had previously been proposed that mobile chemical signals passed between cells coordinate the process by which a lobe in one puzzle cell matches an indentation in its neighbor. However, the model developed by Sapala, Runions et al. does not require such chemical signaling. Instead, mechanical forces and the shape the puzzle cells themselves may transmit this information.

Mechanical forces are known to have important effects on the shape and behavior of cells from other species too. For example, animal cells can develop into different cell types depending on the stiffness of the surface they are placed on. Now that Sapala, Runions et al. have highlighted that plant cell shapes also adapt to mechanical forces, further research is needed to uncover how these forces are sensed.

DOI: https://doi.org/10.7554/eLife.32794.002

divides the plasma membrane into alternating expression domains, with ROP2 in lobes and ROP6 in indentations (*Fu et al., 2009*). These proteins are thought to regulate pavement cell interdigitation through their interactions with RIC proteins, with ROP2 recruiting actin through RIC4 in the lobes, and ROP6 recruiting cortical microtubules through RIC1 and katanin to restrict growth in indentations. Disruptions in the ROP/RIC pathways lead to defects in puzzle cell formation (*Fu, 2002*; *Fu et al., 2005*; *Fu et al., 2009*; *Xu et al., 2010*; *Lin et al., 2013*). Since a lobe in one cell must be matched by an indentation in its neighbor, some manner of extracellular communication is required. The plant hormone auxin has been proposed to act as this signal (*Fu et al., 2005*; *Xu et al., 2010*; *Li et al., 2011*), although recent data call for a re-evaluation of this hypothesis (*Gao et al., 2015*; *Belteton et al., 2018*).

Although these studies have elucidated many of the molecular players involved in puzzle cell patterning, a mechanistic theory is lacking, in part because the function of the puzzle-shape in epidermal cells is unclear (*Bidhendi and Geitmann, 2018*). It has been hypothesized that the interdigitation of the lobes and indentations may strengthen the leaf surface (*Glover, 2000*; *Jacques et al., 2014*; *Sotiriou et al., 2018*), with material sciences studies supporting the plausibility of this idea (*Lee et al., 2000*). Alternatively, puzzle-shaped cells may allow for the correct spacing of the other epidermal cell types, such as guard cells and stomata (*Glover, 2000*). However there is little experimental support for these hypotheses at present. Here we propose a different function for the puzzle shape, that it is an adaptation to a developmental constraint related to the mechanical forces that act on turgid plant cells that reside in the epidermis.

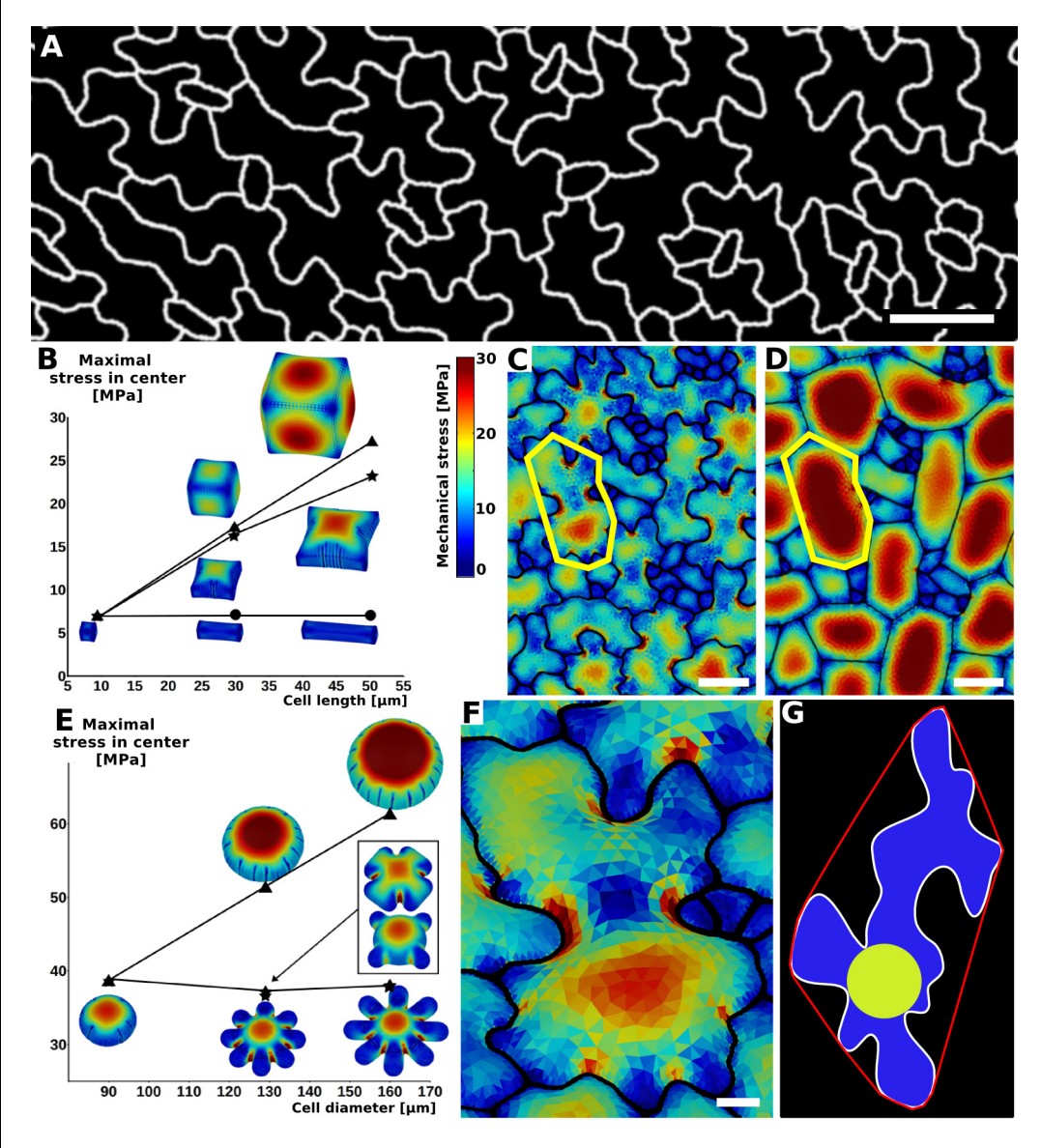

**Figure 1.** Relationship between cell shape and stress. (**A**) Cell contours in adaxial epidermis of an *Arabidopsis thaliana* cotyledon. Small, elliptical cells are stomata. Scale bar: 50 μm. (**B–F**) Cellular stress patterns in finite element method (FEM) simulations. Cell walls have uniform, isotropic material properties (Young's modulus = 300 MPa) and are inflated to the same turgor pressure (5 bar). (**B**) Graph points show stress in cells expanded in one dimension (circles), two dimensions (stars) or three dimensions (triangles). Enlargement in two or more dimensions substantially increases stress in the center of the cell walls. (**C**) Principal stresses generated by turgor in vivo were simulated in a FEM model on a template extracted from confocal data. (**D**) A simplified tissue template using the junctions of the cells in (**C**). The yellow outline marks a corresponding cell in (**C**) and (**D**). Total area and number of cells is the same, however the maximal stress is much lower in the puzzle-shaped cells compared to the more isodiametrically-shaped cells. (**E**) In isolated circular cells, pressure-induced stress increases with diameter (triangles), as was the case in (**B**). Adding lobes, regardless of their length, width or number (inset) does not influence maximal stress in the cell wall in the center (stars). (**F**) A close-up view showing high stress areas that coincide with the center of the large open area of the cell, or indentations that support large open areas. (**G**) Measures used to quantify puzzle cell shape and stress. The largest empty circle (LEC, yellow) that fits inside the cell is a proxy for the maximal stress in the cell wall. The convex hull (red) is the smallest convex shape that contains the cell. The ratio of cell perimeter (white) to the convex hull perimeter gives a measure of how lobed the cell is (termed 'lobeyness'). Scale bars: 50 μm (**C,D**), 10 μm (**F**). Color scale: trace of Cauchy stress tensor in MPa.

DOI: https://doi.org/10.7554/eLife.32794.003

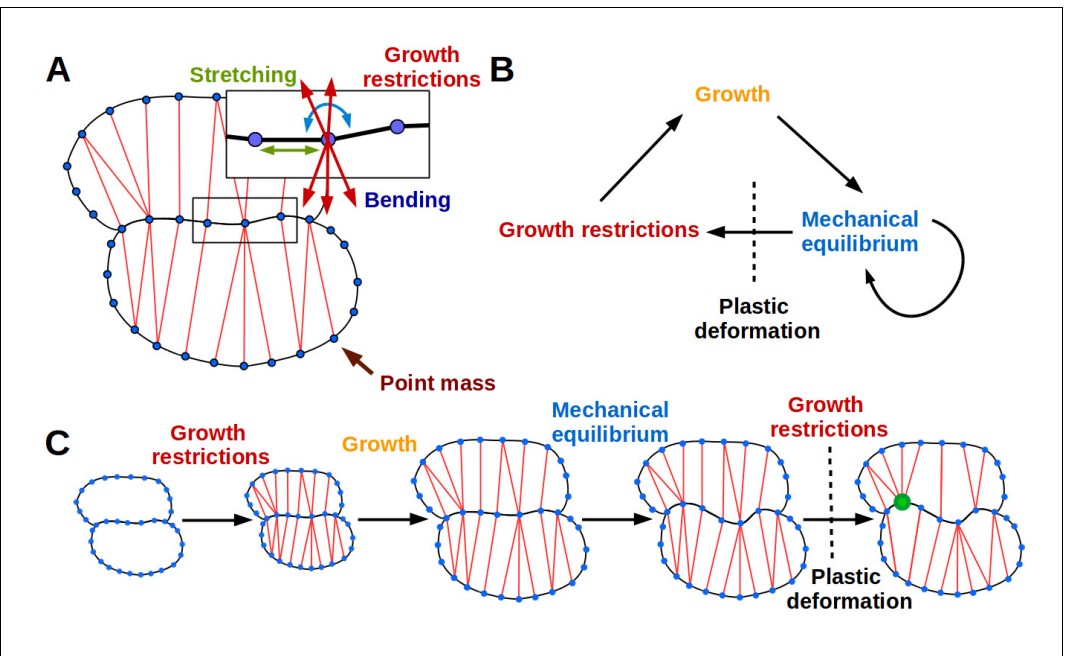

**Figure 2.** The 2D puzzle cell model. (**A**) Mechanical representation of cells. Cell walls are discretized into a sequence of point masses (blue circles) connected by linear wall-segments (black lines). Growth restricting connections (red lines) join point masses across the cell. The forces acting on the point mass are produced by stretching of wall segments and growth restricting connections as well as bending of the cell wall at the mass. (**B–C**) The simulation loop consists of 3 steps (**B**), as depicted for a diagrammatic example in (**C**). Step 1: additional transversal springs (red) are added to the model to represent oriented cell wall stiffening components guided by microtubules connecting opposing sides of the cell. They act like one-sided springs, in that they exert a force when under tension (i.e. stretched beyond their rest length), but are inactive when compressed. This is consistent with the high tensile strength of cellulose. Step 2: the tissue is scaled to simulate growth, which can have a preferred direction (i.e. is anisotropic). Step 3: the network of springs reaches mechanical equilibrium. Transversal springs restrict cell expansion in width, causing cell walls to bend. Before the next iteration, wall springs are relaxed and transversal springs are rearranged to reflect the new shape of cells. Cell shapes emerging in the model are determined by the nature of the assumed tissue growth direction. Note that in (**C**) the deformation of the cell causes the placement of growth restrictions to change during the subsequent iteration, where the green mass at the lobe tip attracts more connections on the convex side and loses connections on the concave side (in line with the model assumption of not restricting growth in concave regions).

DOI: https://doi.org/10.7554/eLife.32794.004

The following figure supplement is available for figure 2:

**Figure supplement 1.** Mechanical properties of the cell wall are simulated using stretching and bending springs.
DOI: https://doi.org/10.7554/eLife.32794.005

Mechanically, plant cells are like small balloons inflated with considerable turgor pressure, up to 10 bar in *Arabidopsis* leaf cells (*Forouzesh et al., 2013*), reaching values up to 50 bar in specialized cells such as stomata (*Franks et al., 2001*). Turgor pressure induces mechanical stress in the cell wall, which is the ratio of the force acting on a cross-section of the material (cell wall) scaled by the area of the material resisting the force. If the wall is made of a homogeneous material, then for a given turgor pressure, cell size and shape provide a good predictor of mechanical stress (*Niklas, 1992*; *Geitmann and Ortega, 2009*), with larger cells subject to more stress than smaller ones (*Bassel et al., 2014*). Although the composition of the cell wall is undoubtedly more complex (reviewed by *Cosgrove, 2005*; *Cosgrove, 2014*), this suggests that cell shape and mechanical stress are intimately connected. Most plant tissues emerge from undifferentiated cells that are initially small and isodiametrically shaped, and subsequently proliferate, differentiate and expand. For epidermal cells composing the outermost cell layer in each organ, minimizing mechanical stress on their walls is likely particularly important as the epidermis limits organ growth and is under tension from

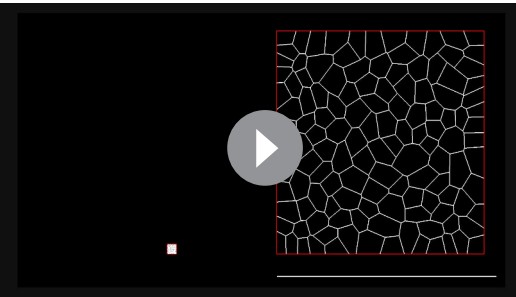 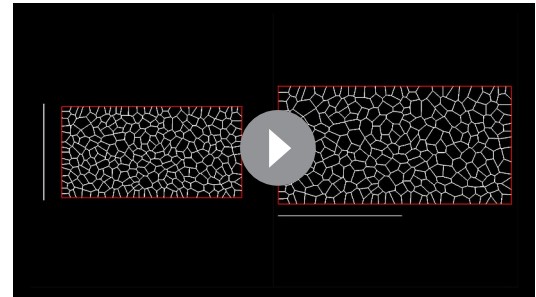

**Video 1.** Simulation of cell shape development in an isotropically expanding tissue. The tissue is shown at two scales: unscaled (Left) and scaled to maintain a constant tissue width (Right). Red lines traversing cell-interiors correspond to active growth restrictions. Scale bar indicates a constant reference length.
DOI: https://doi.org/10.7554/eLife.32794.009

**Video 2.** Simulation of cell shape development in an anisotropically expanding tissue. The tissue is shown at two scales. (Left) Scaled to maintain a constant tissue width. (Right) Scaled so that the largest dimension of the tissue is constant. Scale bars indicate a common constant reference length.
DOI: https://doi.org/10.7554/eLife.32794.010

internal tissues (*Savaldi-Goldstein et al., 2007*; *Kutschera and Niklas, 2007*; *Beauzamy et al., 2015*).

Here we explore the relationship between cell shape and mechanical stress, to understand if mechanical stress is a morphological constraint in shaping epidermal cells. We propose a plausible driver for the creation of the intricate, commonly observed puzzle cell forms by demonstrating that they reduce the forces the cell wall has to withstand. We present computer simulation models that show that actively minimizing force leads to the emergence of the puzzle cell shape, reducing stress and thus potentially lowering the amount of cellulose and other wall material required to maintain mechanical integrity of the cell wall.

## Results

### Cell shape predicts mechanical stress magnitude

Using the Finite Element Method (FEM), we performed simulations on single cells with idealized shapes to explore the effect of cell shape on turgor-induced mechanical stresses (the trace of the Cauchy stress tensor) in the cell wall (*Bassel et al., 2014*). To assess basic relations between cell shape and stress we used uniform, isotropic elastic properties for cell walls, which were assumed to have cell wall thickness of 1 μm, and pressurized the cells to 5 bar (note that this neglects inhomogeneities in the cell wall, as for example observed by *Majda et al., 2017*). Starting with a small cube-shaped cell (10 × 10 × 10 μm) we increased the initial cell size in different dimensions to observe the effect on stress following pressurization. We observed that an increase of cell length in one direction (50 × 10 × 10 μm) does not significantly increase maximal stress in the cell wall (*Figure 1B*). This suggests that anisotropic growth that results in long thin cells is a mechanically advantageous strategy to limit stress magnitude, limiting the wall thickness required to maintain the cell's integrity. Next, we simulated a cell expanded in two directions (50 × 50 × 10 μm) and observed that the maximal stress was much higher. Enlarging the cell in two directions created a large open surface area, causing the cell wall to bulge out in response to turgor pressure, greatly increasing the stress. When the third dimension is enlarged to form a cube (50 × 50 × 50 μm), only a small increase in maximal stress is observed compared to the 50 × 50 × 10 μm case. Thus if a cell must increase its size, an effective way to do it without increasing stress is to elongate along a single axis, instead of expanding in two or three dimensions. Plant organs such as roots, hypocotyls, sepals, many grass leaves and stems grow primarily in one direction and have elongated cells, which would maintain low stress during growth. But how do cells avoid excessive stress if they are part of a tissue that grows in two directions, such as the surface of broad leaves?

Here we propose that the puzzle cell shape, with lobes and indentations, provides a solution to this problem. To test this hypothesis, we began by analyzing the stress in a mechanical model of the

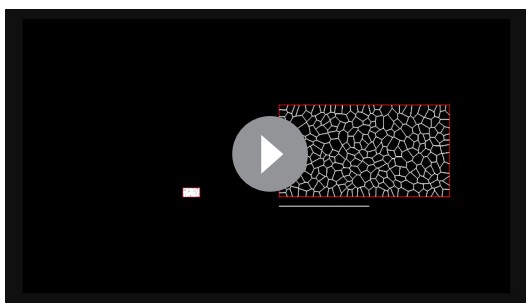

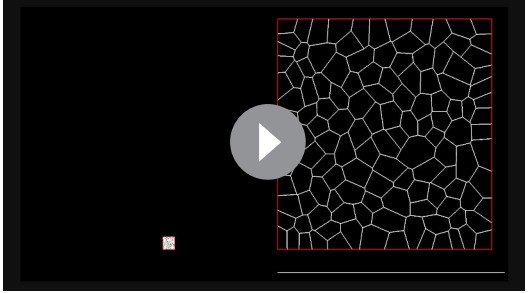

**Video 3.** Simulation of cell shape development in a non-uniformly expanding tissue. Growth anisotropy increases linearly from left to right. The tissue is shown at two scales: unscaled (Left) and scaled to maintain a constant tissue width (Right). Red lines traversing cell interiors correspond to active growth restrictions. Scale bar indicates a constant reference length.
DOI: https://doi.org/10.7554/eLife.32794.011

**Video 4.** Simulation of the development of *spk1*-like cells in an isotropically expanding tissue. The tissue is shown at two scales: unscaled (Left) and scaled to maintain a constant tissue width (Right). Red lines traversing cell-interiors correspond to active growth restrictions. Scale bar indicates a constant reference length of arbitrary value.
DOI: https://doi.org/10.7554/eLife.32794.026

cotyledon epidermis of *Arabidopsis thaliana.* A cellular surface mesh was extracted from confocal images using the image analysis software MorphoGraphX (*Barbier de Reuille et al., 2015*). The mesh was then extruded to form a layer of 3D cells of uniform thickness representing the cotyledon epidermis (*Mosca et al., 2017*). Next, the cells were pressurized, and the stresses visualized (*Figure 1C,F*). In order to examine the effect of lobes on the stress, we created a second template with simplified cell shapes using only the junctions (points shared by three different cells) of the original cells (*Figure 1D*). While the total and average cell area in the original and simplified tissue is the same, the overall stress is much lower in the original (puzzle-shaped) tissue, especially for large cells (*Figure 1C,D*).

Next, we asked how the presence of lobes affects mechanical stress in the cell wall. We computed mechanical stress in idealized circular cells, adding protrusions to simulate the lobes of a puzzle cell. While stress increases with diameter (*Figure 1E*), adding lobes to the original cell does not significantly affect stress in the central part of the cell. Furthermore, increasing lobe length has no impact on stress, although the total volume of the cell increases. Similarly, changing lobe width or number does not affect stress in the cell center (*Figure 1E*, inset). However, there are stress hot spots located between the protrusions, where values appear to be inversely correlated with the width of the protrusion (the distance between the flanks of two consecutive lobes), and increase with the radius of the central part of the cell. This is similar to what we observed in pressurized puzzle cells (*Figure 1C,F*). In both cases, high stress values appear in open areas and in the indentations between protrusions, consistent with previous observations (*Sampathkumar et al., 2014*). In the absence of lobes, the load acting in the middle of the cell is transmitted approximately evenly to the cell contour, whereas in puzzle-shaped cells, the central load is transferred to the area between the lobes, creating stress hot spots in the indentations. The magnitude of stress in the indentations is therefore a direct reflection of the large open areas of the cell that they support, and is thus higher when cells bulge more. Despite the stress hot spots between protrusions, overall stress at both the cell and tissue level is much lower in puzzle shaped cells than in the simplified cell shape template (*Figure 1C,D*).

Following these observations, we propose that the size of the largest empty circle (LEC) that can fit into the cell contour (*Figure 1G*, yellow) can serve as a proxy for mechanical stress magnitude in both puzzle and non-puzzle shaped cells. For long thin cells, such as in roots or stems, the size of the empty circle is the cell diameter, which is known to predict stress for cylindrical cells (*Geitmann and Ortega, 2009*). We hypothesize that in a strongly anisotropically growing organ the plant would make long thin cells, whereas in more isotropic organs puzzle cells would be produced. Counter-intuitively, it is the requirement for isotropic expansion at the tissue scale that drives the irregular shape of puzzle cells.

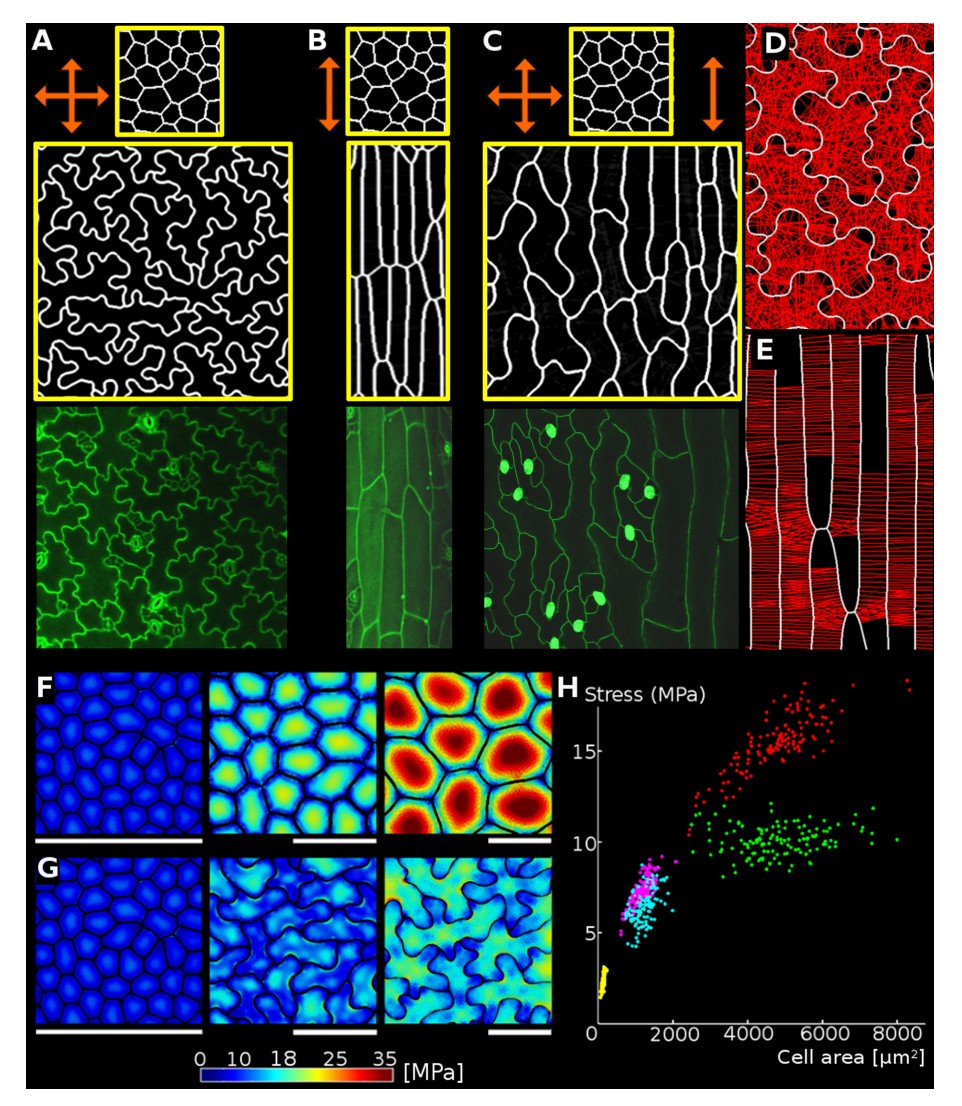

**Figure 3.** Geometric-mechanical model of puzzle cell emergence. (**A**) Starting with meristematic-like cells (top), growing the tissue isotropically, i.e. equally in all directions (arrows), produces puzzle-shaped cells (middle) that resemble cotyledon epidermal cells (bottom). (**B**) Growing the tissue primarily in one direction (anisotropically) results in elongated cells (middle) as observed, for example, in the petiole (bottom). (**C**) A gradient of growth anisotropy (increasing left to right) produces a spatial gradient of cell shapes (middle), as observed between the blade and midrib of a leaf (bottom). (**D–E**) Connections of transversal springs (red) restricting growth in each simulation step in tissues with isotropic (**D**) and anisotropic (**E**) growth. To make connections more apparent, only 50% are visualized. (**F–G**) Cell outlines from 2D models with isotropic growth were used to generate 3D templates for FEM models (growth progresses from left to right, scale bars: 80 μm). (**F**) As the tissue grows, cells lacking transversal springs conserve their original shape. In pressurized cells, mechanical stress increases with the cell size. (**G**) When transversal springs are added, tissue expansion generates lobed cells. (**H**) Average stress in the cell increases with cell area in the polygonal cells (yellow, pink, red), while stress plateaus during tissue grows when cells form lobes (cyan, green). Points of each color represent cells of increasing size, with stresses calculated using the FEM model. Color scale: trace of Cauchy stress tensor in MPa.

DOI: https://doi.org/10.7554/eLife.32794.006

The following figure supplements are available for figure 3:

**Figure supplement 1.** Parameter space exploration for key model parameters.
DOI: https://doi.org/10.7554/eLife.32794.007

**Figure supplement 2.** Varying isotropy for an alternative parameter set.
DOI: https://doi.org/10.7554/eLife.32794.008

## Cell shape measures

To test our hypothesis, a method to quantify the puzzle shape of cells was required. As the epidermis is a surface of relatively uniform thickness, most shape measures applied to puzzle cells consider only the 2D form of cells, and several methods have been developed for this purpose (see *Zhang et al., 2011*; *Wu et al., 2016*, and references therein). A common measure to estimate the complexity of a contour is circularity, indicating how closely a given object resembles a circle. Circularity is calculated using the ratio of the perimeter to the square root of area (*Zhang et al., 2011*; *Majda et al., 2017*). However, it is not suitable for our purposes as both simple, elongated cells and lobed puzzle cells have increased circularity values. Consequently, it cannot be used to reliably distinguish between these cell shapes. Another common approach is to calculate a skeleton based on the cell contour and count its branches (*Le et al., 2006*). Unfortunately skeletonization methods can be very sensitive to small changes in shape (such as the error produced by discretization) making it difficult to robustly quantify the geometric features of cells.

Here we use a method based on the convex hull (*Wu et al., 2016*), the smallest convex shape containing the cell (think of a rubber band surrounding the cell). We define cell *lobeyness* as the perimeter of the cell divided by the perimeter of its convex hull (*Figure 1G*, white and red, respectively). The higher this value, the more lobed the cell is expected to be. The ratio of the cell's convex hull area to that of the cell is another possibility, however we found that for important special cases, such as worm or boomerang-shaped cells, using the area may produce high ratios even when cells do not have significant lobes. The ratio of perimeters (perimeter of the cell/perimeter of its convex hull) is less affected in these cases.

## A mechanistic model of puzzle shape emergence

Cortical microtubules are thought to direct the deposition of cellulose fibrils in the cell wall (*Green, 1962*; *Paredez et al., 2006*). These fibrils stiffen the cell wall, causing growth to be favored in the direction perpendicular to the fibrils (*Suslov and Verbelen, 2006*). Cortical microtubules have also been shown to orient along the maximal direction of tensile stress (*Hejnowicz et al., 2000*; *Hamant et al., 2008*). The fact that growth anisotropy affects cell shape and cell shape affects stress, suggests a feedback mechanism linking cell shape and growth *via* the response of cortical microtubules to mechanical stress directions. This idea is supported by experimental and modeling work showing that predicted stress directions in puzzle cells align with cortical microtubule direction in *Arabidopsis* cotyledons (*Sampathkumar et al., 2014*).

Here we propose a dynamic simulation model of puzzle cell patterning based on the idea that cells can respond to mechanical signals generated by cell geometry. The model focuses on the developmental stage when cells stop dividing and begin to expand. The basic principle behind the model is that as cells grow, stresses gradually increase, and when they reach a threshold level the cell wall is reinforced to resist these stresses. Using simulations on idealized cell templates, we test whether this basic principle is sufficient to generate different cell shapes, depending on the anisotropy of tissue growth. The emerging cell shapes primarily arise from the growth direction imposed at the tissue level that is locally modulated by stress-based growth restriction.

We present the essential aspects of the model here (*Figure 2*), but refer the reader to the Appendix for further details. Cells are represented as polygons (*Figure 2A*), with wall segments between nodes acting like linear springs (*Prusinkiewicz and Lindenmayer, 1990*), and nodes having resistance to bending between adjoining segments (*Matthews, 2002*, *Figure 2—figure supplement 1*). Thereby the model accounts for cell wall thickness and penalizes sharp features, which are usually not observed in nature. A simulation step consists of 3 phases (*Figure 2B,C*). During the first phase, springs are inserted across cells in addition to those defining the cell polygon. These additional springs account for the presence of oriented cell wall stiffening components, such as cellulose microfibrils whose deposition is guided by cortical microtubules (*Paredez et al., 2006*) that are thought to respond to stress (*Hejnowicz et al., 2000*; *Hamant et al., 2008*). The springs also only exert force when they exceed a given target length, related to the LEC, which provides a proxy for stress (*Figure 1*). These connections across the cells introduce growth restrictions into the model, and are placed according to two criteria. First, these springs connect each node to the closest node across the cell falling within a given angle from the normals of the two nodes. Second, connections are inhibited if the the cell wall is convex (see Appendix, Microtubule placement). This facilitates lobe

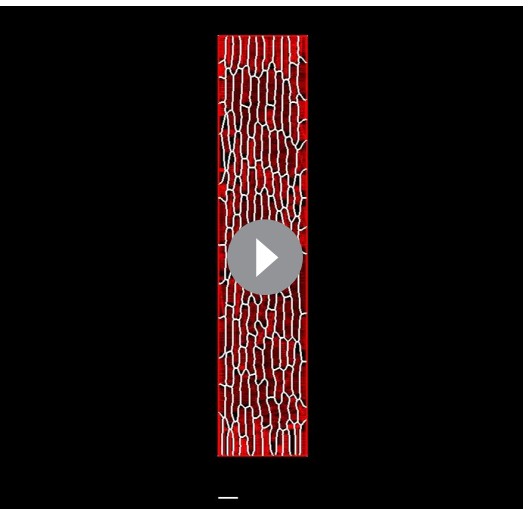

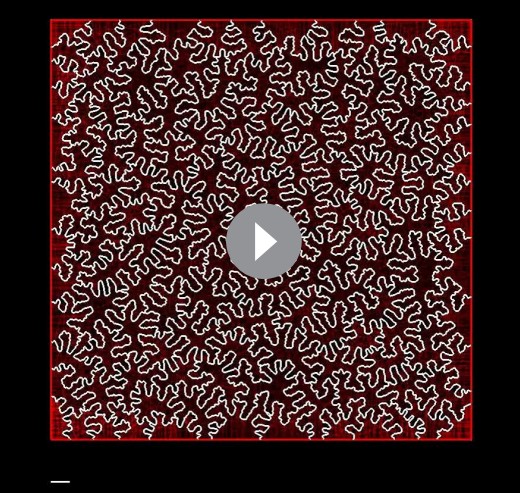

**Video 5.** Simulation results when growth isotropy is varied. Using the wild type isotropic simulation as a reference, growth isotropy (growth in width/growth in height, or $g_x/g_y$ in Appendix) is varied from 50% to 100% of the reference value in regular increments. Successive frames show the final stage of each simulation as isotropy is increased. Scale bars indicate a constant reference length.
DOI: https://doi.org/10.7554/eLife.32794.012

**Video 6.** Simulation results when bending stiffness is varied. Using the wild type isotropic simulation as a reference, bending stiffness ($k_b$ in Appendix) is varied with respect to the reference value from 5% to 25% and then from 25–200% by increments of 25%. Successive frames show the final stage of each simulation as the bending stiffness is increased. Scale bars indicate a constant reference length.
DOI: https://doi.org/10.7554/eLife.32794.013

formation, gives a pattern that both follows the patterns of stress previously reported by *Sampathkumar et al. (2014)* and is consistent with the proposed action of ROP2 in excluding ROP6 from lobes. In the second phase, growth is simulated by displacing the wall segments, based on the specified tissue growth (e.g. isotropically or anisotropically), and relaxing stretched cell-wall springs so that the rest lengths match their actual length. The connections across the cells do not grow. Once placed, their reference length is unaffected by growth and is fixed until the connection is removed. In the third and final phase, a new resting state is found by updating cell shapes to achieve mechanical equilibrium. The next simulation step commences by reassigning microtubule/cellulose connections based on the new cell shape and updating the rest-length of cell wall segments. This highly dynamic arrangement of microtubules is consistent with a similar assumption underlying mechanistic explanations of cell division patterns (*Lloyd, 1991*; *Besson and Dumais, 2011*).

If tissue growth is isotropic, cells quickly approach their target LEC, and connections representing the cellulose and microtubules begin to stretch. Lobes emerge as the indentations (concave regions) attract more connections and protrusions (convex regions) lose connections (*Figure 2C*, *Video 1*). The increased number of connections at indentations is an emergent geometric effect. As the indentation deepens, and its tip becomes more exposed, it becomes the closest node to a larger number of nodes on the opposing cell wall, thus attracting more connections. This is consistent with the findings of *Sampathkumar et al. (2014)*, who detected oriented patterns of mechanical anisotropy with atomic force microscopy, consistent with the proposed directed accumulation of cellulose microfibrils in the indentations of puzzle cells. These connections act as a proxy for the additional stress in the indentations (*Figure 1*). Interestingly, the accumulation of connections in the indentations is consistent with the observed auto-catalytic effect of microtubule bundling in indentations in real pavement cells, *via* induced ROP6/RIC1/katanin-dependent microtubule severing activity (*Lin et al., 2013*; *Sampathkumar et al., 2014*). Conversely, protrusions gradually lose connections as neighboring nodes become closer to opposing portions of the cell wall. This is enhanced by the model assumption that connections cannot be made across the cell to opposite walls from regions that are too convex (i.e. in the lobes, *Figure 3A,D*). If the simulation is performed with anisotropic growth, the cellulose-microtubule connections are never stretched significantly beyond the LEC, and cells

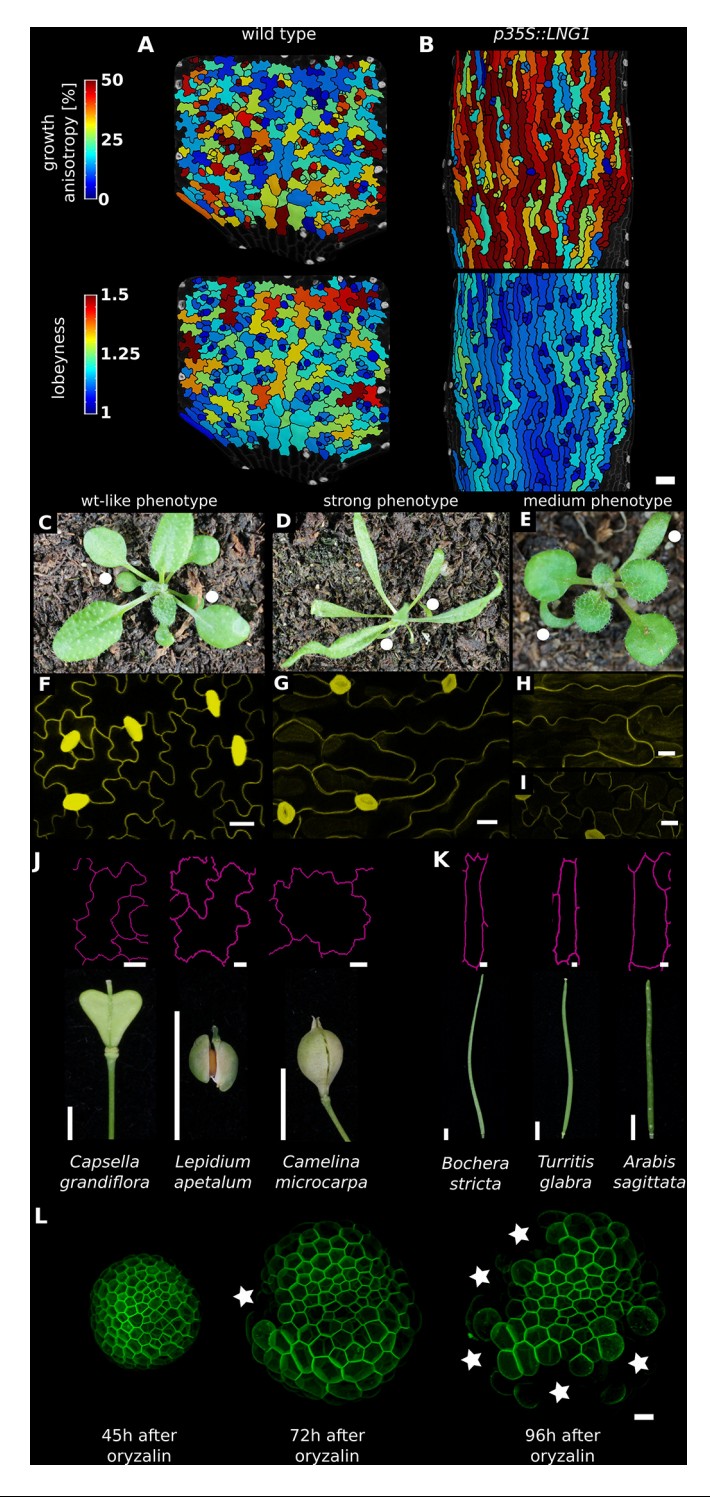

**Figure 4.** Correlation between growth direction and shape on the cell and organ level. (**A-B**) Time-lapse confocal imaging. Pictures were taken every 48 hr and analyzed using MorphoGraphX. The last time point of each series is shown. Growth anisotropy between 2 and 6 days after germination (DAG), calculated as the expansion rate in the direction of maximal growth divided by expansion rate in the direction of minimal growth, and cell lobeyness in wildtype (A) and *p35S::LNG1* (B) cotyledons. The *p35S::LNG1* cotyledon displays more anisotropic growth and less lobed epidermal cells. Scale bars: 50 µm. (**C-E**) *p35S::LNG1* T$_1$ plants with wild type-like phenotype (C, 61/98 plants), strong phenotype with dramatically elongated cotyledons and leaves (D, 16/98 plants) and intermediate phenotype with elongated cotyledons but wt-like leaves (E, 12/98 plants). Cotyledons are marked by white dots.
*Figure 4 continued on next page*

*Figure 4 continued*

The remaining nine obtained plants displayed elongated costyledons and mildly elongated leaves (not shown). (F-I) Confocal images of epidermal cells. Scale bars: 20 μm. (F) shows cells from a leaf in (C), (G) shows cells from a leaf in (D), (H) shows cells from a cotyledon in (E), and (I) shows cells from a leaf in (E). (J-K) Epidermal cell outlines from fruit with more isotropic shapes (silicles, J) and more anisotropic shapes (siliques, K).Fruit images reproduced from Figure 4 and S4 of *Hofhuis et al., 2016*; published under the terms of the Creative Commons Attribution license (http://creativecommons.org/licenses/by/4.0/). Cell outlines reproduced from Figure 2 of *Hofhuis and Hay (2017)*, adapted with permission from John Wiley and Sons. Scale bars: 10 μm for cell outlines, 1 mm for fruit. (L) Depolymerization of cortical microtubules by oryzalin treatment causes cells of NPA-treated meristems to expand without division, ultimately leading to the rupture of the cell wall due to increased mechanical stress. Regions where cells have ruptured (white stars) are primarily located on the flanks of the meristems, where cells are larger. Scale bar: 20 μm.

DOI: https://doi.org/10.7554/eLife.32794.018

The following figure supplement is available for figure 4:

**Figure supplement 1.** Correlation between growth direction and shape on the cell and organ level demonstrated by time-lapse confocal imaging.

DOI: https://doi.org/10.7554/eLife.32794.019

---

simply elongate, and lobes do not emerge (*Figure 3B,E*). In other words, stress-based activation of connections induces indentations, coinciding with locations of ROP6 activity, which necessarily generate incipient lobes in adjacent portions of the cell-wall where ROP2 is localized, accentuating their outgrowth. Thus, although phrased in geometric terms, our model is consistent with both the antagonistic local molecular interactions of ROP2-ROP6 and the stress-based feedbacks proposed by *Sampathkumar et al. (2014)*.

The main parameters of the model are the stiffness of the cell walls and the cellulose-microtubule connection springs, the target LEC, the angle within which connections can be made, and the convexity criteria for attachment to the opposing wall (see Appendix Table 1 for parameter values). To examine the contribution of growth distribution to cell shape we varied growth anisotropy while all other model parameters remained constant (*Figure 3A–C*). In this case, the emergence of puzzle vs. elongated cells depends only on the anisotropy of growth at the tissue scale, with puzzle cells appearing for isotropic growth, and elongated cells for anisotropic growth (*Video 1*, *Video 2*). If the growth specified has a gradient of anisotropy at the tissue scale, a gradient of cell shapes from elongated to lobed is produced (*Video 3*). Similar gradients in cell shape are seen in *A. thaliana* leaves, where elongated cells cover the anisotropically growing midrib, whereas lobed cells adorn the adjacent isotropically growing leaf blade (*Figure 3C*).

To explore the effect of model parameters on cell morphology we performed a parameter space exploration using the simulation with isotropic growth as a reference (*Figure 3A*). We varied isotropy within a range of 40–100% of the reference value and all other parameters within a range of at least 25–200% (*Figure 3—figure supplement 1*, *Videos 5–10*). This exploration showed that when growth is anisotropic, there is no strict 'threshold' for the onset of lobing, but rather it is a continuous characteristic. This feature is preserved when the initial template and additional parameters are varied (*Figure 3—figure supplement 2*). Parameter variation also demonstrates that the model can generate a diverse range of plausible cell shapes, similar to those observed in nature (e.g. *Figure 3—figure supplement 2B* 60% isotropy; which are reminiscent of epidermal cells in maize leaves).

To validate the model, we confirmed in a FEM analysis that limiting the size of the LEC by creating lobes during growth reduces the cellular stress (*Figure 3F–G*). This causes the maximum stress in simulated tissues to plateau, greatly reducing it compared to isodiametric cells of the same size (*Figure 3H*). The model thus illustrates how a mechanism actively limiting the mechanical stress of cells by restricting large open areas (LECs) can lead to the formation of puzzle-shape cells in the context of isotropic growth.

The model relies on cell-autonomous mechanical restriction of indentations through controlled cellulose deposition and does not require cell-cell signaling molecules to synchronize the indentations in one cell with the protrusions of its neighbor. Nonetheless, synergies exist between the mechanical and biochemical control of cell morphogenesis. In particular, the ROP6 in the indentation of one cell must coincide with ROP2 in the corresponding lobe of the neighboring cell.

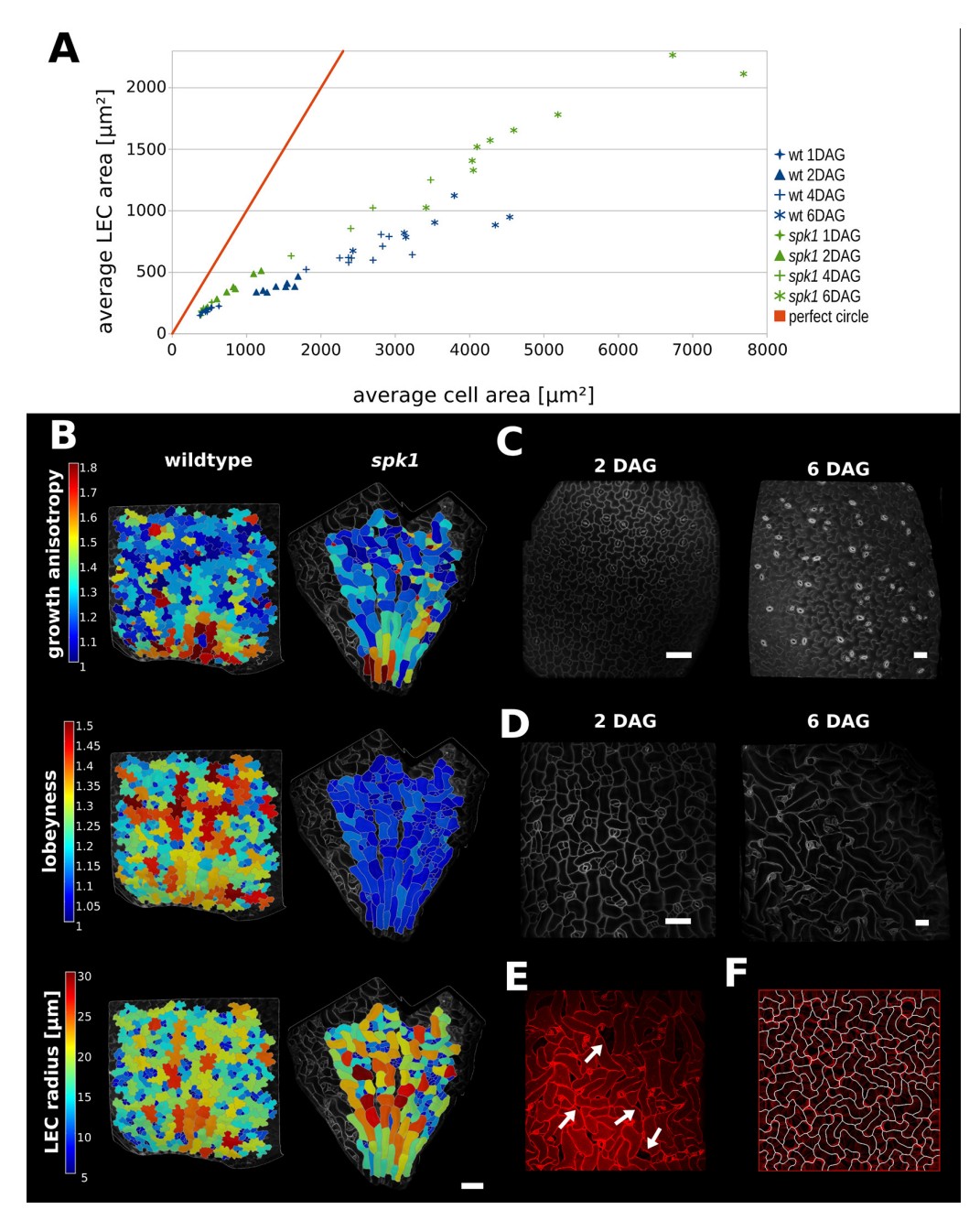

**Figure 5.** Characterization of *spike1* mutant. (**A**) Average LEC area of wild type and *spk1* cotyledons vs. average cell area. The red line represents the theoretical case of a perfectly circular cell. In this case cell area and LEC area increase at the same rate. For the cell area and the LEC area analysis we considered the average values for the largest 20% of segmented cells in order to avoid bias stemming from the much smaller cells of the stomatal lineage, which due to their small size would not need to regulate their LEC. For average values for each point, including sample size and SE, see *Figure 5—source data 1*. (**B**) Time-lapse data on wild type and *spk1* cotyledons. Plants were imaged twice in 48 hr intervals. Heat maps are displayed on the last time points. Scale bar: 100 µm. (**C and D**) Examples of cell shapes in the experiment shown in (**A**). Scale bars: 50 µm. (**C**) Wild type. (**D**) *spk1*. (**E**) Confocal image of *spk1* cotyledon, 8 DAG. Note the gaps between cells and ruptured stomata that typify *spk1* phenotype (arrows indicate several examples). (**F**) Model result with placement of transverse connections in lobe tips combined with parameter changes to account for defects in ROP-mediated cytoskeletal rearrangement (see main text).

DOI: https://doi.org/10.7554/eLife.32794.020

*Figure 5 continued on next page*

*Figure 5 continued*

The following source data and figure supplements are available for figure 5:

**Source data 1.** Average cell area and LEC area for wt and *spk1* cotyledon time-course.
DOI: https://doi.org/10.7554/eLife.32794.024

**Source data 2.** Mean average cell area for wild type and *spk1* cotyledon cells (20% largest segmented cells for each sample, averaged), displayed in *Figure 5—figure supplement 3*.
DOI: https://doi.org/10.7554/eLife.32794.025

**Figure supplement 1.** Comparison of growing wt and *spk1* cotyledons.
DOI: https://doi.org/10.7554/eLife.32794.021

**Figure supplement 2.** Cellular stress patterns in *spike1* cells.
DOI: https://doi.org/10.7554/eLife.32794.022

**Figure supplement 3.** Mean average cell area for wild type and *spk1* cells.
DOI: https://doi.org/10.7554/eLife.32794.023

---

Although mobile biochemical signals cannot be ruled out (*Xu et al., 2010*), our model predicts that this signal could be passed through the geometry of the cells *via* its effect on stress patterns or geometry, with indentations attracting microtubule-cellulose deposition and ROP6, and lobes suppressing microtubules *via* the cell-autonomous co-repression of ROP2 and ROP6.

## Isotropic tissue growth is correlated with puzzle-shaped cell formation

Our model predicts that puzzle cells should appear when cells stop dividing and tissue growth is not primarily in one direction. To test this prediction experimentally, we performed time-lapse confocal imaging on cotyledons (n = 3 time-lapse series), which have a blade of roughly isodiametric shape, growing from 2 to 4 days after germination (DAG). Epidermal cells of *Arabidopis thaliana* cotyledons begin to acquire a puzzle-shaped morphology roughly 2 DAG, whereas the organ achieves its characteristic round shape at approximately 3 DAG, long before reaching its final size (*Zhang et al., 2011*). We used MorphoGraphX (*Barbier de Reuille et al., 2015*) to extract growth rates and directions, and these results confirm that the overall growth of cotyledons is isotropic as suggested by its round shape. To examine the correlation between growth anisotropy and lobeyness we pooled the data from the final time-point of our time-lapse series. We then extracted the largest 100 cells from this set (i.e. those most likely to be affected by the stress-minimizing mechanism) and found a significant correlation between growth anisotropy and lobeyness (Pearson correlation coefficient r = −0.46, p=0.6 × 10⁻⁶). This supports our hypothesis that growth anisotropy and lobeyness are inversely related in the isotropically growing cotyledons of *Arabidopsis* (see also *Figure 4A*, *Figure 4—figure supplement 1C*).

In contrast to cotyledons, the *Arabidopsis* sepal is an elongated organ with epidermal cells that are either small and relatively isodiametric in shape, or large and elongated. Sepals initiate from a band of cells in the floral meristem, undergoing strongly anisotropic growth (*Hervieux et al., 2016*) which produces giant cells that are far less lobed than those of the cotyledon (compare *Figure 4—figure supplement 1A and C*). Thus growth isotopy and final organ shape correlate with lobeyness in these two organs.

Next we examined cases where genetic modifications changed growth anisotropy and overall organ shape. Sepals of the *ftsh4* mutant show increased variability of organ shape (*Hong et al., 2016*). In some samples, the growth is more isotropic than wild type, and cells of more isodiametrically shaped organs exhibit decreased growth anisotropy and increased lobeyness, and start to become puzzle shaped (compare *Figure 4—figure supplement 1A and B*). The shift from anisotropic to isotropic growth in the sepal is thus correlated with a shift from elongated giant cells to puzzle-shaped cells.

The opposite change in growth anisotropy and organ shape can be seen in plants overexpressing the *LONGIFOLIA1* (*TRM2, LNG1*) gene. This causes an elongated cell and organ phenotype in *A. thaliana* cotyledons and leaves (*Lee et al., 2006*; *Drevensek et al., 2012*), consistent with effects of a related protein in rice grains (*Wang et al., 2015*). We created transgenic plants where *LNG1* is overexpressed under the CaMV 35S promoter (*p35S::LNG1*). Our T₁ lines had phenotypes ranging from highly elongated cotyledons and leaves to wild type (*Figure 4C–E*). Plants with the elongated phenotype grew more anisotropically than wild type and had epidermal cells with reduced lobeyness

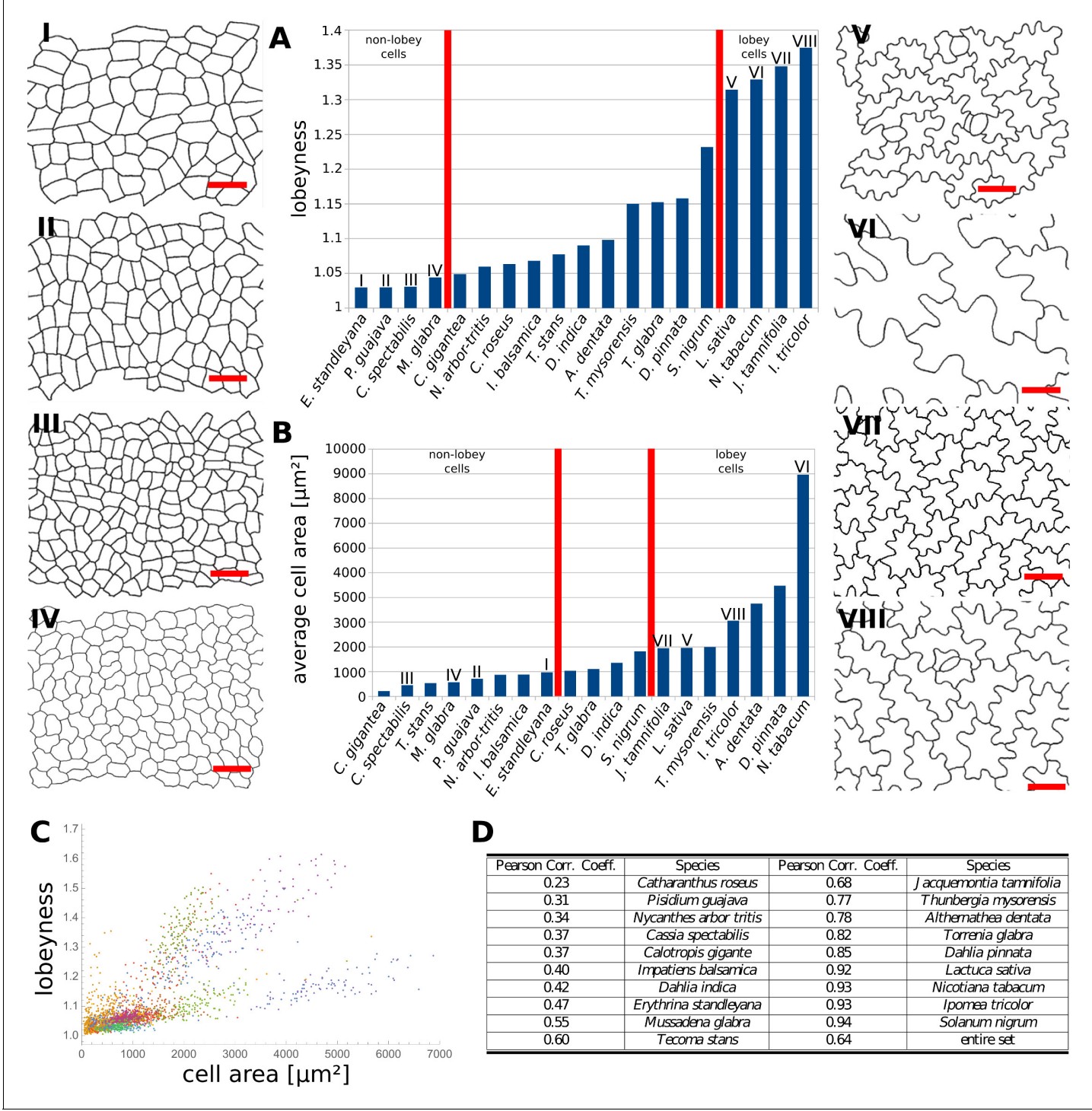

**Figure 6.** Multi-species cell shape analysis. (A) Average cell lobeyness. (B) Average cell area. (I-VIII) Pictures of leaf epidermal cells of species corresponding to numbering in (A) and (B), numbered by the order of appearance in (A). Scale bars, 50 µm. (C) A plot of lobeyness vs. area for cells of all species pooled together. Each color symbolizes one species. (D) Pearson correlation coefficients between lobeyness and cell area for each species and for all cells pooled together (entire set). Note that in all cases a positive correlation between lobeyness and cell area is observed (correlation coefficient is greater than 0).

DOI: https://doi.org/10.7554/eLife.32794.027

The following source data is available for figure 6:

**Source data 1.** Average cell area and lobeyness for all studied species.

DOI: https://doi.org/10.7554/eLife.32794.028

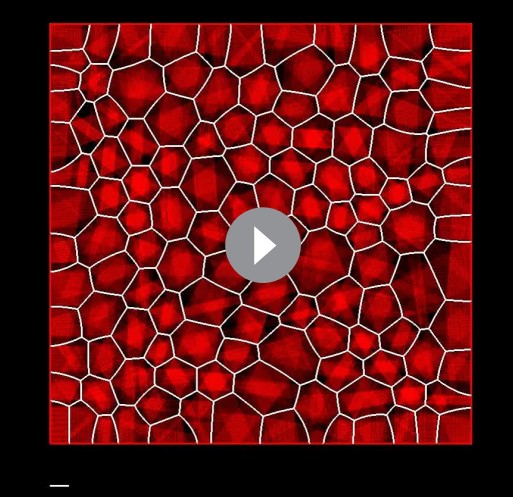

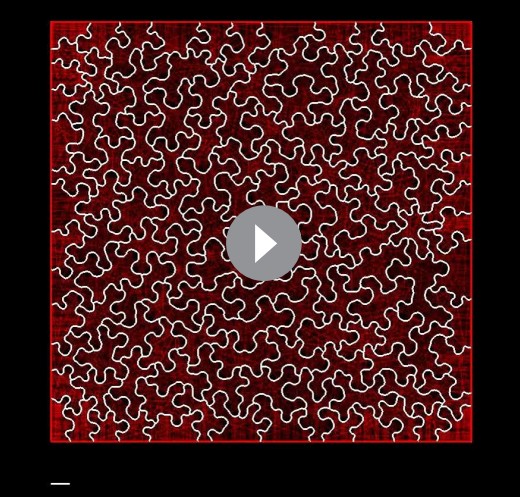

**Video 7.** Simulation results when cellulose stiffness is varied. Using the wild type isotropic simulation as a reference, cellulose connections stiffness ($k_m$ in Appendix) is varied with respect to the reference value from 0–200% by increments of 25%. Successive frames show the final stage of each simulation as the cellulose connections stiffness is increased. Scale bars indicate a constant reference length.
DOI: https://doi.org/10.7554/eLife.32794.014

**Video 8.** Simulation results when stretching stiffness is varied. Using the wild type isotropic simulation as a reference, stretching stiffness ($k_s$ in Appendix) is varied with respect to the reference value from 25–200% by increments of 25%. Successive frames show the final stage of each simulation as the stretching stiffness is increased. Scale bars indicate a constant reference length.
DOI: https://doi.org/10.7554/eLife.32794.015

(n = 3 time-lapse series for each genotype, *Figure 4A–B and F–I*). Thus the change in growth and organ shape from isodiametric to elongated correlated with a decrease in cell lobeyness.

To further test the generality of the correlation between organ and cell shape, we examined fruit epidermal cells in a sample of 21 species from the Brassicaceae family (full dataset shown in *Hofhuis and Hay, 2017*). These fruit pods were either elongated siliques or short, rounded silicles and we only observed puzzle-shaped cells in silicles, not in siliques (*Figure 4J,K*). This strict correspondence between fruit shape and puzzle-shaped epidermal cells fits the prediction of our model that puzzle shapes are required to allow cells to enlarge in isotropically growing tissues, but are not required in elongated organs.

## Lobeyness allows cells to increase their size while avoiding excessive mechanical stress

Our model predicts that plant cells regulate their shape to prevent their LEC, a proxy for stress, from becoming too large. To test this hypothesis, we imaged cells of young cotyledons at different stages of growth and tracked changes in cell and LEC area. We reasoned that if the cell area increases faster than LEC area, cells must have a mechanism to maintain a low LEC radius. We imaged 1, 2, 4 and 6 DAG seedlings, as within this time window we could qualitatively observe the most dramatic increase in cellular lobeyness. In the epidermal cells of 2 DAG seedlings, lobes were small or absent in most cells, while 6 days after germination most cells were puzzle-shaped (*Figure 5C*). For each time point we imaged up to 10 plants and segmented several hundred cells from each plant using MorphoGraphX. We then pooled all cells from each timepoint and calculated average cell area and LEC area for the largest 20% of cells (*Figure 5A*, *Figure 5—source data 1*).

We compared these values to the case where cells are perfectly isodiametric (i.e. circles) so that the cell area and LEC area are equal (*Figure 5A*, red line). Our results show that as the cotyledon grows, the ratio of average LEC area to average cell area increases slower than when the cell is circular. Consequently, as organ development progresses, cell area increases faster than LEC area, consistent with the idea that increased lobeyness allows surface area to increase faster than the magnitude of stress (*Figure 5A*, blue signs).

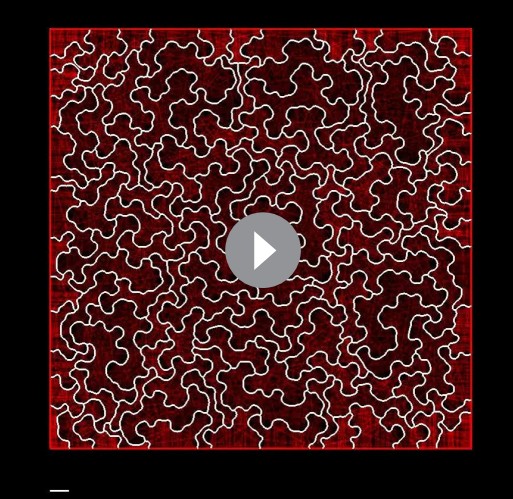
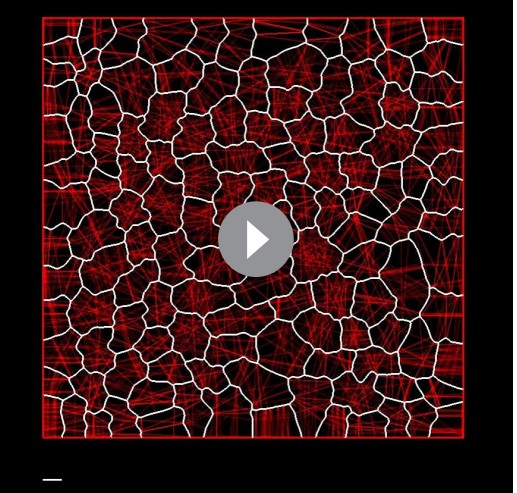

**Video 9.** Simulation results when target LEC is varied. Using the wild type isotropic simulation as a reference, target LEC ($min_{micro}$ in Appendix) is varied with respect to the reference value from 0–200% by increments of 25%. Successive frames show the final stage of each simulation as the target LEC is increased. Scale bars indicate a constant reference length.
DOI: https://doi.org/10.7554/eLife.32794.016

**Video 10.** Simulation results when normal angle is varied. Using the wild type isotropic simulation as a reference, normal angle ($\theta_{micro}$ in Appendix) is varied with respect to the reference value from 25–200% by increments of 25%. Successive frames show the final stage of each simulation as the normal angle is increased. Scale bars indicate a constant reference length.
DOI: https://doi.org/10.7554/eLife.32794.017

## Experimental evidence that mechanical stress needs to be managed

Our model and experiments show that a mechanism, likely cortical-microtubule dependent, generates puzzle shapes to limit stress in large cells when tissue growth is isotropic. It is commonly observed that the periclinal cell walls slightly bulge out in healthy, turgid cells. However, if stress is indeed a developmental constraint, then when cells grow isotropically without this mechanism, they should bulge excessively, reach their rupture point and burst. The shoot apex of *Arabidopsis* grows isotropically in areas without lateral organs (*Kwiatkowska and Dumais, 2003*; *Kierzkowski et al., 2012*), with the cells presumably managing their mechanical stress by employing cell division to remain small. In plants grown with auxin transport inhibitor 1-N-naphtylphtalamic acid (NPA), the shoot apex is unable to produce lateral organs, and is uniformly covered in small rapidly dividing cells of isodiametric shape (*Reinhardt et al., 2000*). Treating these meristems with oryzalin, a chemical compound that depolymerizes cortical microtubules, blocks cell division and anisotropic growth restriction, preventing the formation of puzzle shapes. It has been shown in *Arabidopsis* hypocotyls that oryzalin treatment changes the trajectory of cellulose microfibril-producing molecules (CESA), as there is no organized cortical microtubule array to follow, but does not appear to change the rate of cellulose production (*Chan et al., 2010*). As such, although oryzalin makes cell walls mechanically isotropic by preventing the directionally organized deposition of cellulose, it does not necessarily reduce the overall deposition of cellulose, although this cannot be precluded. Cells of shoot apices in these conditions do not divide, but continue to grow developing large, isodiametric shapes that tend to balloon out (*Hamant et al., 2008*; *Corson et al., 2009*; *Grandjean, 2004*).

After treating naked meristems of NPA-grown seedlings with oryzalin (five biological replicates), 20 displayed full microtubule depolymerization following oryzalin treatment (as assessed by the absence of cell division). In those 20 samples, we could see cell bursting in the latest time points of 13 samples, out of which 10 displayed bursting cells located in the flank of the meristem, where cells were substantially larger (*Figure 4L*). Although it cannot be completely excluded that these lateral cells, under these experimental conditions, have different wall properties, the most parsimonious explanation is that their cell walls could not withstand the increasing mechanical stress induced by

the isotropic expansion. This provides direct experimental support for the proposition that large iso-diametrically shaped cells are not viable due to the high stresses on their walls.

## A strategy for when lobes cannot be formed

Previous reports have shown that lobe formation in pavement cells is compromised in *spike1* (*spk1*) mutants (*Qiu et al., 2002*). The SPIKE1 protein is a guanine nucleotide exchange factor (GEF) and is required for the production of the active, GTP-bound form of ROP proteins molecular switches that deliver signals to downstream components. SPIKE1 regulates actin polymerization *via* WAVE and ARP2/3 complexes (*Basu et al., 2008*). Furthermore, it activates ROP2, ROP4 and ROP6, thereby promoting isotropic cell expansion (*Ren et al., 2016*). Mutant plants have a number of severe phenotypes including reduced trichome branching, altered organ shape and increased sensitivity to low humidity environments. Epidermal cells of *spk1* plants have altered shape, with lobes either small or absent, and compared to wild type, their overall cell shape is much less complicated. Furthermore, the epidermis suffers from defects in cell-cell adhesion, which have been reported to result in gaps between cells that are clearly visible in the cotyledon epidermis from approximately 5 DAG on (*Qiu et al., 2002*; *Ren et al., 2016*). It has been reported that *spk1* cotyledons are narrower, but not longer than wild type cotyledons (*Qiu et al., 2002*) and *spk1* petals display an increase in growth anisotropy at late stages of development, after the general shape of the organ has been established (*Ren et al., 2016*).

In *spk1,* epidermal cells of the cotyledons do not have puzzle-shaped forms (*Figure 5B,D,E*, *Figure 5—figure supplement 1B*), even though the tissue growth isotropy is similar to wild type plants (n = 3 time-lapse series, *Figure 4B*, *Figure 5—figure supplement 1A* – cellular growth anisotropy shows a small statistical difference in that *spk1* grows more anisotropically than wild type). Given our hypothesis that lobes function to reduce mechanical stress (LEC size) during isotropic growth, we tested if LEC was higher in the simple-shaped cells of *spk1* mutant than in the puzzle-shaped cells of wild type. In our time-lapse experiment, even though lobeyness is greatly reduced in *spk1*, LEC radius in the final time point is comparable to wild type (*Figure 5B*, *Figure 5—figure supplement 1C*). Our FEM simulations revealed that cellular stresses in wild type and *spk1* cells are similar and scale with LEC (compare *Figure 1C* and *Figure 5—figure supplement 2*). We also performed the same analysis as for wild type, imaging *spk1* cotyledons 1, 2, 4 and 6 DAG and measuring cell area and LEC area (*Figure 5C,D*). This revealed a similar trend to that observed in the wild type, with cell area increasing faster than LEC area (*Figure 5A*) during the course of organ development. At the same time, mean average cell area in *spk1* remained similar or lower than in wild type until 6 DAG (*Figure 5—figure supplement 3*). Cells of the *spk1* mutant keep LEC low and overall organ growth remains isotropic. Similar LEC size in mutant and wild type suggests that LEC acts as a threshold for stress based cell shape modification. However, instead of forming lobes, the *spk1* cells themselves seem to interdigitate generating worm-like shapes. It is possible that this strategy is insufficient, as holes appear between cells in the growing epidermis of cotyledons (*Figure 5E*), which may be due to increased mechanical stress. However, since holes are already present in cotyledons at 1 DAG it is more likely that they result from direct disruption of the molecular process regulating cellular adhesion, such as actin-driven pectin delivery to cell walls, causing defects prior to the stress-based shape patterning where the final cell shape is established. The *spk1* mutant is unable to make lobes because it fails to activate ROPs which interact with effector proteins to mediate cytoskeletal rearrangements and cell shape (*Basu et al., 2008*). In our model framework, ROP2 activity would preclude connections where walls have high curvature, thus preventing connections from penetrating lobes. Apart from removing this assumption from the model, we increased the stiffness of connections and the cell-wall, and decreased the frequency at which connections were reset, to account for defects in ROP-mediated cytoskeletal rearrangement. These three changes to the initial simulation allowed us to reproduce the *spk1* phenotype (*Figure 5F*, *Video 4*). This suggests that creating interdigitated worm-shaped cells provides an alternate strategy to cover an isotropically growing tissue, although possibly not as efficient in reducing stress as lobe formation.

## Cell shape and size across species

Our data indicates that the stress control mechanism we propose is conserved between various organs in *A. thaliana*, and within the fruit of Brassicaceae (*Figure 4J,K*). This raises the question as

to how broadly this mechanism is conserved, with large cell size and isotropic growth correlating with puzzle-shaped cells. Under this assumption, two geometric strategies are possible for cell expansion in isotropically growing organs without requiring excessively thick walls: (1) keeping cell size small by frequent divisions or (2) creating larger, puzzle-shaped cells. We measured cell area, LEC area and lobeyness in the adaxial epidermis of 19 unrelated plant species including trees, shrubs and herbs. A statistical analysis revealed that there was a positive correlation between cell size and lobeyness for each species (*Figure 6A–C*). Species with the largest average lobeyness also tended to have the largest cells (and vice-versa, *Figure 6A–B*). For average values of lobeyness and cell area of each species (including sample size), see *Figure 6—source data 1*. Pearson correlation coefficients ranged from 0.23 for *Catharantus roseus* to 0.94 for *Solanum nigrum* (*Figure 6D*). When pooling cells of all species together, the Pearson correlation coefficient was 0.64. Lobe formation is therefore more likely to be observed in big cells rather than small cells, which is intuitive if one considers cell division (where cell size remains low) as an alternative strategy to limit LEC size and cell wall stress. This suggests our hypothesis, that plants create puzzle-shaped cells in order to reduce stress in large isotropically growing cells, may be conserved among many plant species.

## Discussion

We propose that the puzzle shaped cells seen in the epidermis of many plant species emerge from a mechanism that evolved to limit mechanical stress in tissues that grow isotropically, such as the epidermis of leaves and cotyledons. FEM analysis of 3D pressurized cells shows that cell shape influences the direction and magnitude of mechanical stress exerted on the cell wall. When an epidermal cell becomes large in two directions (i.e. has a large open area), stress is greatly increased. In stems, roots and siliques, growth is strongly anisotropic, and cells can simply elongate. This is, however, not possible for isodiametric organs such as broad leaves, cotyledons and silicle fruit pods. We propose that puzzle-shaped cells in the epidermis of more isodiametric plant organs provide a means to avoid large open areas in the cell and the high stresses that they induce. Since turgor pressure inside cells is high, minimizing mechanical stress by shape regulation may be a way of reducing the resources required to reinforce the cell wall and at the same time maintaining its structural integrity during growth.

Although it is possible that the interlocking puzzle-shaped cells have a role in strengthening the epidermal cell layer (*Glover, 2000*; *Jacques et al., 2014*), our experimental data shows that growth anisotropy correlates with cell shape, a prediction that does not appear to readily follow from this alternative hypothesis. Organs displaying isotropic planar growth have puzzle shaped cells while anisotropically growing organs have more elongated cells with fewer lobes. Genetic perturbations that modify growth anisotropy in either direction result in the predicted changes in cell shape. In the *Arabidopsis thaliana* cotyledon, a *p35S::LNG1* overexpression line changes growth from isotropic to anisotropic, and cell shapes become more elongated with fewer lobes. Conversely, in sepals of the *ftsh4* mutant, growth is switched from anisotropic to more isotropic, and the elongated giant cells become more puzzle shaped. Our hypothesis is also consistent with the mild lobing of pavement cells in grass leaves, which often have strongly anisotropic growth (*Sylvester et al., 2001*)

Although studies often focus on anisotropic growth at the cellular level when analyzing puzzle cell development (*Armour et al., 2015*), our hypothesis suggests that isotropic growth at the tissue level is a primary driver of cell shape. As a tissue grows the stress increases, and microtubules align to direct cellulose deposition to resist the stress. This causes small indentations in the cell, which transfers more stress to them, further recruiting microtubules and more cellulose deposition. The process generates a local activation feedback of cell shape on growth *via* the mechanical stresses that are induced by that shape.

A geometric-mechanical simulation model of these processes confirms that the hypothesis is plausible, and the model is able to produce puzzle-shaped cells from a few simple assumptions. Since we use a geometric proxy for stress (LEC), the possibility that the cells sense their geometry through chemical means is also compatible with the model. Our model explains the gradual emergence of lobed cells from polygons resembling meristematic cells, providing an explanation for the till now enigmatic morphogenesis of these distinctive cells. The model suggests that the main driver of the complex puzzle shape comes from the restriction of growth in the indentations, rather than the promotion of growth in the protrusions. It also predicts that the puzzle cell shape is triggered by

isotropic growth, and that puzzle cell morphogenesis may not require any signaling molecules to coordinate a protrusion in one cell with the corresponding indentation of its neighbor. Nonetheless, the model does not preclude a role for inter-cellular signaling, which could reinforce patterns produced by geometry sensing or facilitate the initial steps of lobing (*Majda et al., 2017*).

The mechanism also predicts that spatial differences in cell wall material properties, corresponding to lobes and indentations, should appear in periclinal cell walls as organized cellulose distributions appear, consistent with observations that cellulose-bundles accumulate in high-stress indentations (*Sampathkumar et al., 2014*). Spatial differences in stiffness corresponding to incipient lobes and indentations have recently been measured in cross-sections of anticlinal cell walls (*Majda et al., 2017*). Although the direction of the measurements (z-direction) is not explicitly represented in our model, it is nevertheless consistent with the idea that material properties in adjacent cell walls would be expected to be different in the lobe side vs the indentation side. The modeling results of *Majda et al., 2017* suggest these mechanical differences drive the formation of small lobes and indentations when the anticlinal walls are placed under tension by turgor pressure. Although this cannot explain deep lobes and indentations or lobes on lobes that emerge in maturing puzzle cells, the idea that both anticlinal cell walls and periclinal cell walls play a coordinated role in puzzle shape morphogenesis (*Belteton et al., 2018*) is appealing, and warrants further study using FEM models of pavement cell morphogenesis that represent the entire 3D geometry of cells (c.f. *Bidhendi and Geitmann, 2018*).

Our model is also consistent with the functions attributed to the main molecular players that have been reported to influence puzzle cell formation, the ROP family of GTP-ases. The elaboration of puzzle shape is influenced by two antagonistic molecular pathways. On the convex side (protrusion), ROP2 and ROP4 inactivate the microtubule-associated protein RIC1, thereby suppressing the formation of microtubule arrays, and activate RIC4 which enhances the assembly of actin microfibrils. This was proposed to result in growth promotion (*Fu et al., 2005*). On the concave side (indentation), ROP6 activates RIC1 and katanin, promoting the formation of bundled microtubule arrays that restrict growth (*Fu et al., 2009*; *Lin et al., 2013*). The theory of coordinated outgrowth and restriction has struggled to provide an explanation as to how protrusions and indentations are coordinated between cells. As ROP2 and ROP6 were believed to be activated by auxin (*Xu et al., 2010*), it was suggested that auxin could act as the mobile signal underlying in this coordination (reviewed in *Chen et al., 2015*) via ABP1, however this scenario seems unlikely given recent genetic evidence that ABP1 alone does not have quantifiable effects on auxin response (*Gao et al., 2015*). Nonetheless, our model is also consistent with the idea that ROP6 is a part of the stress sensing mechanism, and that stress (or strain) is the trigger for localized ROP6 accumulation. Currently, the molecular mechanism for how stress (or strain) could be sensed and its relationship to the ROPs is unknown, although microtubules have been proposed to respond to stress *in planta* (*Hejnowicz et al., 2000*; *Hamant et al., 2008*). Since stress is closely related to shape in pressurized plant cells, a curvature sensing mechanism could be involved (*Higaki et al., 2016*), similar to that proposed for villi patterning during gut morphogenesis (*Shyer et al., 2015*). Simulations have shown that a ROP2-ROP6 co-repression network can indeed partition a cell in discrete domains of ROP2 and ROP6 expression (*Abley et al., 2013*). Our data suggest that this intracellular partitioning network works in concert with a mechanical or geometric signal, transmitted by the shape of the cell itself.

## Materials and methods

### Key resources table

| Reagent type (species) or resource | Designation | Source or reference | Identifiers | Additional information |
|---|---|---|---|---|
| gene | spk1 | Nottingham Arabidopsis Stock Centre | SALK_125206 | |
| gene | ftsh4-5 | *Hong et al., 2016* | | |
| genetic reagent | p35S::LNG1 | this paper | | Vector obtained using gateway cloning, transformed into Col-0 plants by Agrobacterium-mediated floral dipping |

*Continued on next page*

*Continued*

| Reagent type (species) or resource | Designation | Source or reference | Identifiers | Additional information |
|---|---|---|---|---|
| genetic reagent | pUBQ10::myrYFP | *Hervieux et al., 2016* | | |
| recombinant DNA reagent | LNG1CDS | this paper | | Full-length CDS of LONGIFOLIA1 gene, PCR amplified |
| recombinant DNA reagent | pENTR/D-TOPO | Invitrogen | | |
| recombinant DNA reagent | pK7WG2 | *Karimi et al., 2002* | | |
| genetic reagent | p35S::LTI6b-GFP | *Cutler et al., 2000* | | |
| other | N-(1-naphtyl) phtalamic acid (NPA) | *Hamant et al., 2008* | | |
| other | oryzalin | *Hamant et al., 2008* | | |
| software, algorithm | VVE | *Smith et al., 2003* | | www.algorithmicbotany.org |
| software, algorithm | MorphoGraphX | *Barbier de Reuille et al., 2015* | | www.MorphoGraphX.org |

## Live imaging of cotyledons

Plantlets were grown on 1/2 MS medium in long day conditions as previously described in *Vlad et al. (2014)*. Young cotyledons (2–6 days after germination, DAG) were imaged using the Leica SP8 microscope with 20x (HCX APO, numerical aperture 0.8) and 40x (HCX APO, n.a. 0.5) long working distance, water immersion objectives. Col-0 and *p35S::LNG1* plants contained a plasma membrane-localized fluorescent marker *pUBQ10::myrYFP* previously described in *Hervieux et al., 2016* and fluorescent signal was collected from 519 to 550 nm emission spectrum using 514 nm laser for excitation. Sepals were imaged as previously described in *Hong et al., 2016* and *Hervieux et al., 2016*. For *spk1* plants and corresponding Col-0 controls, cell walls were stained with propidium iodide and fluorescent signal was collected from 605 to 644 nm emission spectrum using 488 nm laser for excitation. *spk1* homozygous mutant cotyledons were chosen for time-lapse imaging 2 DAG based on their shape, which was more elongated compared to wild type cotyledons of comparable age.

## Creating transgenic lines

The LNG1 gene full-length CDS was PCR amplified and cloned into the pENTR/D-TOPO vector (Invitrogen) as described in the manual, using primer pair 5'-CACCATGTCGGCGAAGCTTTTGTATAACT-3' and 5'-GAACATAAGAAAGGGGTTCAGAGA-3'. The resultant vector was LR recombined into the gateway vector pK7WG2 (*Karimi et al., 2002*) to generate the final construct *p35S::LNG1*. The intermediate and final constructs were verified by sequencing. The *p35S::LNG1* construct was individually transformed into Col-0 plants by *Agrobacterium*-mediated floral dipping. T1 seeds were sown on Kanamycin-containing medium and transferred into soil approximately 2 weeks after germination.

## Analysis of fruit and exocarp cell shape

Fruit shape was classified as an elongated silique or a silicle (if the length was less than three times the width of the fruit) for 21 species in the Brassicaceae family. Exocarp cells were stained with propidium iodide, imaged by CLSM (as described in section 'Live imaging of cotyledons') and cell outlines extracted using MorphoGraphX.

## Time-course imaging of cotyledons

Arabidopsis seeds were sown on a 1/2 MS medium. 1, 2, 4 and 6 days after germination (DAG) 5–10 seedlings were taken out of the medium and imaged. Confocal stacks were processed in MorphoGraphX (*Barbier de Reuille et al., 2015*). Cell area and LEC radius were calculated for each cell in each sample. For average values displayed in *Figure 5A* (scatter plot), only the largest 20% of cells in each sample were considered, to eliminate stomata and small cells in the stomatal lineage.

## Pharmacological treatment

The *p35S::LTI6b-GFP Arabidopsis* lines have been described previously (*Cutler et al., 2000*) and were grown in tall petri dishes on a solid custom-made Duchefa 'Arabidopsis' medium (DU0742.0025, Duchefa Biochemie) supplemented with 10 μM of NPA (N-(1-naphthyl) phthalamic acid) as described in *Hamant et al. (2008)*. As soon as naked inflorescences had formed, the plants were transferred to a medium without inhibitor. First images (T = 0 hr) were taken 1 day after the plants were taken off the drug. The samples were then immersed for 3 hr in 20 μg/ml oryzalin at T0h, T24h and T48h, as described in *Hamant et al. (2008)*. Images were acquired using a Leica SP8 confocal microscope. GFP excitation was performed using a 488 nm solid-state laser and fluorescence was detected at 495–535 nm.

## Comparison of the distributions of cellular quantities between WT and *spike1*

We employed the Kolmogorov-Smirnov (K-S) test to statistically test if the distributions of growth anisotropy, lobeyness and LEC radius between WT and spk1 were the same. We used heat maps created in MorphoGraphX on data displayed in *Figure 5B* (final time point) to extract the values for each segmented cell. In the K-S test, the cumulative distribution of the corresponding quantity is first constructed as in *Figure 5—figure supplement 1*. The test statistic in the K-S test is the maximum (vertical) distance between the two cumulative distributions from WT and spk1. A large vertical distance signifies that the null hypothesis, i.e., the distributions of WT and spk1 are the same, is more likely to be rejected. The significance level of 0.05 is used in our analysis and we statistically conclude that the two distributions are different if the p-value<0.05.

## *spike1* genotyping

The seeds of a heterozygous spk1 T-DNA insertion line (SALK 125206) were purchased from Nottingham Arabidopsis Stock Centre. Segregating individuals were genotyped according to instructions provided by the SALK institute (http://signal.salk.edu/) using primers 5'- GATTTCAGTCTC TCACCGCAG-3' and 5'-ATGGTCGACTCCACATTTCTG-3' for detecting individuals with no T-DNA insertion and primers 5'-ATTTTGCCGATTTCGGAAC-3' (recommended by SALK) and 5'-ATGG TCGACTCCACATTTCTG-3' for detecting individuals containing the T-DNA insertion (mutant plants).

## Multi-species leaf cell shape analysis

Leaf surface impressions were taken from the adaxial side using transparent nail enamel (Revlon). The impressions were viewed under the differential interference contrast (DIC) mode of an Olympus BX52a upright microscope (Olympus, Japan) and imaged using a CapturePro CCD camera (Jenoptik, Germany). Images were loaded into MorphoGraphX and cell outlines were projected on a flat (2D) mesh. The mesh was segmented, cell area, lobeyness and LEC radius were calculated for all segmented cells.

## Lobeyness and largest empty circle

The Lobeyness and Largest Empty Circle (LEC) measures are calculated using custom plugins developed for MorphoGraphX (*Barbier de Reuille et al., 2015*). These measures are applied to 2D cell contours, obtained by projecting each 3D cell-contour extracted using MorphoGraphX on a local plane. For this purpose, the plane minimizing the loss of variance following projection is used. This plane is obtained from Principal Component Analysis (PCA) of the contour points, and is defined as the plane orthogonal to the third principal component (i.e. the direction of minimal variance) passing through the mean of the contour. Lobeyness captures the deviation of 2D cell contours from the convex polygonal forms typical of young undifferentiated cells. The measure is computed by taking the ratio of the cell's perimeter to that of its convex hull (the smallest convex shape containing the cell), and is the inverse of the convexity measure used in *Wu et al., 2016*. Lobeyness takes a value of 1 for convex shapes and increases with contour complexity. This provides a translation, scale and rotation invariant measure of contour complexity and overall pavement cell lobation. The LEC for each cell is computed using the Delaunay triangulation of the contour positions. The cell contour defines a bounded region of the plane, and the largest empty circle within this region must be either

the circumscribed circle of a triangle in the Delaunay triangulation, or a point on the boundary (*Toussaint, 1983*). Thus, the LEC for each cell is calculated by:

1. Computing the Delaunay triangulation of the projected cell-contour.
2. Calculating the radii for the circumscribing circle of each triangle within the cell.
3. Returning the radius of the largest circle.

As the cell-contours extracted from MorphoGraphX are densely sampled compared to the size of cells, the possibility that the largest empty space corresponds to a point on the boundary is ignored.

## Acknowledgements

We thank Arezki Boudaoud for discussions. Funding for this research is gratefully acknowledged from the Swiss National Science Foundation SystemsX.ch iPhD grant 2010/073 to RSS, the Bundesministerium für Bildung und Forschung grants 031A492 and 031A494 to RSS, Human Frontiers Science Program grant RGP0008/2013 to AHKR, CBL, OH, and RSS, the European Commission through a Marie Skłodowska-Curie individual fellowship (Horizon 2020, 703886) to AR, Natural Sciences and Engineering Research Council of Canada Discovery Grant RGPIN-2014–05325 to PP, European Research Council grant ERC-2013-CoG-615739 ''MechanoDevo'' to OH, and a core grant from the Max Planck Society to MT.

## Additional information

### Funding

| Funder | Grant reference number | Author |
| --- | --- | --- |
| Swiss National Science Foundation | SystemsX.ch iPhD grant 2010/073 | Richard S Smith |
| Bundesministerium für Bildung und Forschung | 031A492 | Richard S Smith |
| Human Frontier Science Program | RGP0008/2013 | Chun-Biu Li Olivier Hamant Adrienne HK Roeder Richard S Smith |
| European Commission | Marie Skłodowska-Curie individual fellowship (Horizon 2020 703886) | Adam Runions |
| Natural Science and Engineering Research Council of Canada | Discovery Grant RGPIN-2014-05325 | Przemyslaw Prusinkiewicz |
| European Research Council | ERC-2013-CoG-615739 'MechanoDevo' | Olivier Hamant |
| Max Planck Society | Core grant and open-access funding | Miltos Tsiantis Richard S Smith |
| Bundesministerium für Bildung und Forschung | 031A494 | Richard S Smith |

The funders had no role in study design, data collection and interpretation, or the decision to submit the work for publication.

### Author contributions

Aleksandra Sapala, Conceptualization, Formal analysis, Validation, Investigation, Writing—original draft, Writing—review and editing; Adam Runions, Conceptualization, Formal analysis, Validation, Investigation, Visualization, Writing—original draft, Writing—review and editing; Anne-Lise Routier-Kierzkowska, Formal analysis, Investigation, Writing—original draft, Writing—review and editing; Mainak Das Gupta, Formal analysis, Investigation, Writing—review and editing; Lilan Hong, Hugo Hofhuis, Stéphane Verger, Investigation; Gabriella Mosca, Software, Investigation, Methodology; Chun-Biu Li, Data curation, Formal analysis, Validation, Investigation, Methodology; Angela Hay,

Data curation, Formal analysis, Supervision, Validation, Methodology; Olivier Hamant, Adrienne HK Roeder, Miltos Tsiantis, Przemyslaw Prusinkiewicz, Supervision, Writing—review and editing; Richard S Smith, Conceptualization, Supervision, Funding acquisition, Writing—original draft, Writing—review and editing

### Author ORCIDs

Aleksandra Sapala (ID) http://orcid.org/0000-0003-3640-398X
Adam Runions (ID) http://orcid.org/0000-0002-7758-7423
Stéphane Verger (ID) http://orcid.org/0000-0003-3643-3978
Chun-Biu Li (ID) http://orcid.org/0000-0001-8009-6265
Olivier Hamant (ID) http://orcid.org/0000-0001-6906-6620
Adrienne HK Roeder (ID) http://orcid.org/0000-0001-6685-2984
Richard S Smith (ID) http://orcid.org/0000-0001-9220-0787

### Decision letter and Author response

Decision letter https://doi.org/10.7554/eLife.32794.034
Author response https://doi.org/10.7554/eLife.32794.035

## Additional files

### Supplementary files

• Transparent reporting form
DOI: https://doi.org/10.7554/eLife.32794.029

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

## Appendix 1

DOI: https://doi.org/10.7554/eLife.32794.030

# Finite element method (FEM) simulations

## FEM code implementation

FEM simulations were performed using 2D membrane elements embedded in 3D space. Each biological cell is a closed surface tessellated with triangles which have a mathematically assigned thickness able to change during the simulation to compensate for the Poisson effect (plane stress hypothesis is adopted). Pressure is assigned as an external load that is applied normal to the surface of the deforming mesh. The material law adopted for the simulations is the hyperelastic Saint-Venant Kirchhoff model, which extends the linear Hooke's law to the non-linear deformation regime. For the computation of the mechanical equilibrium, we use a time-stepping method previously described in *Mosca et al., 2017* .

The stress measures displayed in the main text figures are the sum of the principal in-plane Cauchy stresses (the out-of plane principal stress is zero by hypothesis). Those are computed element-wise with the usual transformation law from the Second-Piola Kirchoff stress tensor and by exploiting the plane stress hypothesis to extrapolate the strain in the cross-sectional direction of the wall. The stress matrix obtained is diagonalized analytically in order to obtain the principal stress values and the principal stress directions.

## Generation of templates for FEM simulations

All the templates were generated with our in-house built software MorphoGraphX (*Barbier de Reuille et al., 2015*). We created cuboid cells in the following dimensions (in $\mu$m): 10 × 10×10, 30 × 30×30 and 50 × 50×50 for the isotropic shape; 30 × 30×10 and 50 × 50×10 for the flat cells; 30 × 10×10 and 50 × 10×10 for long cells. The cell surface was divided into isoceles triangles with 1.77 $\mu$m sides.

To create idealized rounded cell shapes, we generated implicit surfaces with a fixed distance from given seed points. For example, a single seed would generate a spherical surface. Cylindrical cells were generated from groups of seed points densely packed along a line, each end-point generating the cells spherical caps. We used a fixed radius of 15 $\mu$m to generate all surfaces. Flat cells were created using seed points arranged in disks of different diameters: 60, 100 and 130 $\mu$m, corresponding to a maximal cell dimension of 90, 130 and 160 $\mu$m respectively, for a total cell depth of 30 $\mu$m. Branched cells were generated by pulling out points from 4 or 8 selected locations from the 90 $\mu$m disk. The points positions were chosen so that the maximal distance between branches would be 130 or 160 $\mu$m. The cell surfaces were triangulated using marching cubes of 3 $\mu$m, creating triangles of sides between 1.5–2.6 $\mu$m, with a total number of vertices ranging from 8000 for the small disk to 20,000 for the largest star shape. For tissue templates, a first triangular mesh was extracted either from confocal images or from snapshots of the simulations of pavement cell formation. The mesh was then segmented into cells as described in Barbier de Reuille et al. (*Barbier de Reuille et al., 2015*) and flattened into a single plane. Next the mesh was simplified to keep only vertices on the cell outlines, with a distance of about 2 $\mu$m between neighbor vertices. This simplified mesh was used as a basis to create the three dimensional cells. The flat space between the cell outlines was filled with a triangular mesh with edges approximately 4.5 $\mu$m in length (maximally 15 $\mu$m$^2$ in area). Triangles on the cell border connected to the cell outlines were approximately half this size. Vertices of the cell outlines were regularly spaced 2 $\mu$m apart. This ensured a finer mesh resolution at the cell contours, where stress gradients are high. This triangulation was performed once and used to generate mirror triangles on the bottom and top faces. Side faces were generated by extruding the cell outlines along the Z axis for a total depth of 30 $\mu$m and discretized with regular triangles 5 $\mu$m in height. The vertical faces of neighboring cells touch each other and are rigidly connected in real plant

tissues, therefore their vertices were merged to create a single face separating each pair of neighbor cells. Finally the tissue mesh was smoothed five times using the Smooth Mesh plugin in MorphoGraphX to make cells slightly bulge out. After smoothing the vertical walls connecting cells were about 18 $\mu$m high (final triangle height about 4.5 $\mu$m), while at the center the cells were 30 $\mu$m deep. Smoothing the mesh also produced a progressive change in triangle size between cell centers and their boundaries (*Appendix 1—figure 1*).

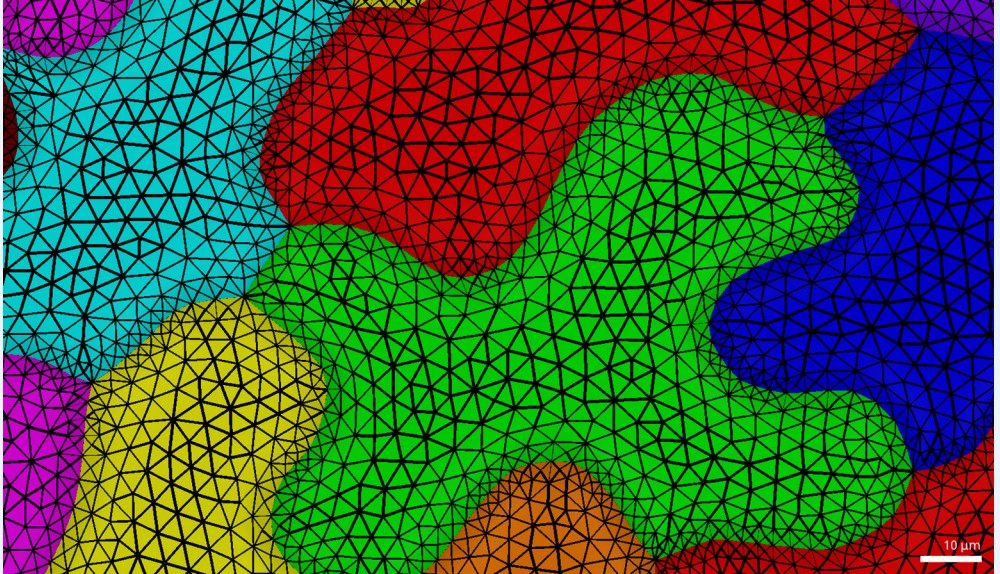

**Appendix 1—figure 1.** Pavement cell triangulation used in FEM simulations (*Figure 1*). Note the smooth gradient of triangle sizes from the cell center towards its margin.
DOI: https://doi.org/10.7554/eLife.32794.031

## Parameters for FEM simulations

All the cellular templates were pressurized to 0.5 MPa. The same parameters were used for all simulations: Young's modulus: 300 MPa, Poisson's ratio: 0.3, Cell wall thickness: 2 $\mu$m for walls shared between two cells (vertical walls), 1 $\mu$m for the remainder.

# Model of pavement cell morphogenesis

## Overview

Models of pavement cell morphogenesis were implemented in C ++ using the VVe simulation framework (an extension of Vertex-Vertex systems; *Smith et al., 2003*). VVe provides a framework for simulating growing 2D cellular-tissues. Cell walls were represented as a series of point-masses connected by springs (*Figure 2*). Each point-mass stored its position, and each spring a length representing its unstressed or *rest length*. Masses and springs residing on the tissue boundary were marked, and constrained to move with the boundary as it grew, inducing tension throughout the cellular network. Additional springs, representing mechanical constraints resulting from the deposition of cellulose and microtubules in response to stress, were dynamically introduced and updated during simulation. The simulation loop and details of the models implementation are provided below. Parameter values for all simulations reported in the main text are provided in *Appendix 1—table 1*. A C ++ implementation of the simulations reported in the main text is available upon request.

**Appendix 1—table 1.** Parameters for simulations of puzzle cell morphogenesis. All simulations use the parameter values specified for isotropic growth (fourth column) unless otherwise specified.

| Parameter | | | Simulation | | | |
|---|---|---|---|---|---|---|
| Name | Symbol | Text reference | Isotropic growth | Anisotropic growth | Growth gradient | *spk1* mutant† |
| Initial width and height | | Overview | 8.5 × 8.5 | 17 × 8.5 | 17 × 8.5 | |
| Cell number | | Overview | 128 | 256 | 256 | |
| Edge subdivision threshold | $Th_{sub}$ | Overview | 0.5 | | | |
| Growth step | $\Delta t$ | Growth | 0.0075 | | | 0.015 |
| RERG along y-axis | $g_y$ | *Equation 1* | 1.25 | 1.0 | 1.5 | 1.5 |
| RERG along x-axis (uniform growth) | $g_x$ | *Equation 1* | 1.25 | 0.35 | NA | 1.5 |
| RERG along x-axis (non-uniform growth) | $[g_{min}, g_{max}]$ | *Equation 2* | | | [0.8,1.35] | |
| Maximum cell wall angle | $\theta_{micro}$ | Microtubule placement | $\frac{\pi}{4}$ | | | |
| Cell wall tensional elastic modulus | $k_s$ | *Equation 7* | 0.4 | | | 2.0 |
| Bending spring constant | $k_b$ | *Equation 8-10* | 0.02 | | | 0.12 |
| Microtubular tensional elastic modulus | $k_m$ | *Equation 11* | 0.75 | | | 2.0 |
| Minimum active length | $min_{micro}$ | *Equation 11* | 3 | | | |
| Maximum active length | $max_{micro}$ | *Equation 11* | 300 | | | |

† Microtubular connections can terminate in recessed portions of the cell-wall (i.e. condition 3 in Microtubule placement is ignored).

DOI: https://doi.org/10.7554/eLife.32794.032

## Simulation execution and initialization

Simulations were initialized with a rectangular tissue, created by successively dividing a rectangular cell (see *Appendix 1—table 1* for initialization parameters). To provide a realistic cellular template cell division was performed using the shortest-wall rule, where the division wall was shortened following division (i.e. *pinched*; see *Nakielski, 2000*). To allow for a more detailed representation of cell geometry, wall segments exceeding a given threshold ($Th_{sub}$) were subdivided. Finally, the rest length for each wall segment was initialized to coincide with its current length.

Following initialization, simulations proceeded according to the simulation loop depicted in *Figure 2B,C*. At the start of each iteration, springs were added across the cell, representing the mechanical reinforcement of the periclinal wall resulting from deposition of cellulose and microtubules (Microtubule placement). Next, the cellular network was grown by displacing cell-wall segments and relaxing cell-wall springs according to the specified growth (Growth). A new rest state was then found by updating the positions of masses to achieve mechanical equilibrium (Mechanical simulation). Once mechanical equilibrium was found, the rest lengths of stretched cell-wall springs were set to their current lengths, to emulate the plastic deformation of cell walls. At the end of the simulation loop, cell wall segments exceeding the threshold $Th_{sub}$ were subdivided to maintain a smooth approximation of the wall.

## Microtubule placement

To simulate mechanical anisotropy of the periclinal wall, resulting from the deposition of cellulose and microtubules that reinforce the cell in response to stress (*Sampathkumar et al., 2014*), additional growth restricting connections (also termed microtubular connections) were introduced spanning the cell contour. These connections were reset every iteration of

the simulation. In every cell, each mass $m_i$ was considered as the starting point for a connection within the cell. A connection between masses $m_i$ and $m_j$ was introduced provided $m_j$ was the closest mass to $m_i$ meeting the following conditions:

1. The mass $m_j$ was visible from $m_i$ (i.e. the line connecting the masses did not pass through the cell wall).
2. The line connecting $m_i$ and $m_j$ was within $\theta_{micro}$ of the inward facing cell-wall normal at $m_i$ and $m_j$. This precluded the formation of oblique and unnatural connections.
3. The mass $m_i$ was closer to $m_j$ than its neighboring masses (i.e. the masses adjacent to $m_i$). This precluded the introduction of connections in recessed portions of the cell wall, consistent with the proposed role of ROP2 in excluding ROP6 from lobes (**Fu et al., 2005**; **Fu et al., 2009**).

If no such mass $m_j$ existed then no connection starting at mass $m_i$ was created. The growth restricting connections acted like springs (see Mechanical simulation), but did not resist compression (i.e. a force was only generated when under tension). The rest length $r_{ij}$ for the growth-restricting connections between masses $m_i$ and $m_j$ with positions $\mathbf{P_i}$ and $\mathbf{P_j}$ was initialized to $||\mathbf{P_i} - \mathbf{P_j}||$, the current distance between the two masses.

## Growth

During the growth step the positions of masses and rest lengths of springs were updated. In the simulations presented in the main text, three growth patterns were considered:

1. uniform isotropic (i.e. uniform scaling),
2. uniform anisotropic (i.e. non-uniform scaling), and
3. non-uniform growth with anisotropy varying linearly with spatial position.

In all cases, growth was described as a deformation that maps the position $\mathbf{P_i^t}$ of mass $m_i$ at time $t$ to its position at time $t + \Delta t$ (i.e. $\mathbf{P_i^{t+\Delta t}}$, where $\Delta t$ is the time step for growth). As we specify growth as a deformation, the grown rest lengths of wall springs were determined by the length of the corresponding wall-segment following growth.

Uniform growth (cases 1 and 2 above) was specified using constant factors $g_x$ and $g_y$ to respectively indicate the Relative Elementary Rate of Growth (RERG, see **Richards and Kavanagh, 1943**) along the x and y axes. Change in the position of $\mathbf{P_i^t} = (x_i^t, y_i^t)$ was then

$$\mathbf{P_i^{t+\Delta t}} = \mathbf{P_i^t} + (g_x x_i^t, g_y y_i^t)\Delta t. \tag{1}$$

According to this equation, growth was isotropic when $g_x = g_y$ and anisotropic when $g_x \neq g_y$, with principal growth directions aligned with the coordinate axes.

To simulate non-uniform growth where anisotropy varies linearly within the simulation domain (case 3), the relative elementary rate of growth along the x-axis (denoted $g_x(x)$) was assumed to vary with x-position. Growth along the y-axis was uniform, and $y_i$ still changed according to **Equation 1**. Accordingly, the function $g_x(x)$ took the following form

$$g_x(x) = (g_{max} - g_{min})\frac{(x_1^t - x)}{(x_1^t - x_0^t)} + g_{min}, \tag{2}$$

where $g_{max}$ and $g_{min}$ were the maximum and minimum growth rates, $x_0^t$ was the x-position of the left boundary of the tissue at time $t$, and $x_1^t$ was the x-position of the right boundary of the tissue at time $t$. This equation took the value $g_{max}$ at $x_0^t$, and linearly decreased to $g_{min}$ at $x_1^t$. The change of position of $x_i^t$ was then

$$x_i^{t+\Delta t} = x_i^t + \Delta t \int_{x_0^t}^{x_i^t} g_x(s)ds. \tag{3}$$

Due to the simple form of **Equation 2**, the preceding integral has a closed form which can be used to compute $x_i^{t+\Delta t}$.

## Mechanical simulation

Changes in cell geometry due to growth were simulated by representing the mechanical properties of cells using a mass-spring system and finding mechanical equilibrium (**Prusinkiewicz et al., 1990**, Chapter 7; **Mosca et al., 2018 [in press]**). Equilibrium is achieved when the force $\mathbf{F_i}$ acting on each mass $m_i$ is zero. We found mechanical equilibrium by evolving the velocity $\mathbf{v_i}$ and position $\mathbf{P_i}$ of the $i^{th}$ mass $m_i$ as follows

$$\frac{d\mathbf{v_i}}{dt} = \frac{\mathbf{F_i} - b\mathbf{v_i}}{w}, \tag{4}$$

$$\frac{d\mathbf{P_i}}{dt} = \mathbf{v_i}, \tag{5}$$

where $w$ was the mass of $m_i$ (assumed to be equal for all masses) and $b$ was the damping coefficient (required to eliminate oscillations). In simulations, equilibrium was found using the forward Euler method with adaptive time-stepping, which was iterated until the maximum force acting on each mass was small (i.e. $\|\mathbf{F_i}\| < \epsilon$ for all $i$). The positions of masses residing on the boundary of the simulation domain were fixed.

The total force $\mathbf{F_i}$ that acted on mass $i$ was related to the mechanical representation of cells (**Figure 2A**), and was given by

$$\mathbf{F_i} = \mathbf{F_s} + \mathbf{F_b} + \mathbf{F_{micro}} \tag{6}$$

where $\mathbf{F_s}$ was the force produced by stretching of adjacent wall segments, $\mathbf{F_b}$ was the force produced by bending of the wall at $m_i$, and $\mathbf{F_{micro}}$ was the force resulting from microtubule directed growth restrictions.

The force $\mathbf{F_s}$ produced by stretching of the cell wall was the sum of forces $\mathbf{F_{s_j}}$ due to stretching of the springs connecting $m_i$ to neighoring masses $m_j$ (**Figure 2—figure supplement 1**):

$$\mathbf{F_{s_j}} = -k_s \frac{l_{ij} - r_{ij}}{r_{ij}} \cdot \frac{\mathbf{P_i} - \mathbf{P_j}}{l_{ij}}, \tag{7}$$

where $k_s$ was the tensional elastic modulus, $r_{ij}$ was the rest length of the spring connecting $m_i$ and $m_j$, and $l_{ij} = \|\mathbf{P_i} - \mathbf{P_j}\|$ was the actual length of the spring. To better approximate the properties of cells, compressed walls (i.e. $l_{ij} < r_{ij}$) produced significantly smaller restoring forces. This was implemented by using $\frac{1}{8}k_s$ in place of $k_s$ for compressed cell-wall springs.

The force $\mathbf{F_b}$ produced by bending of the cell wall was generated by bending springs (**Matthews, 2002**), which were placed in each cell $c$ at every mass $m_n$. This spring resisted bending of the cell wall by exerting a force on $m_n$ and the adjacent masses $m_o$ and $m_r$ (**Figure 2—figure supplement 1**). The force acting on $m_o$ was

$$\mathbf{F_{b_c}^o} = k_b \frac{(\theta_c - \theta)}{\|\mathbf{P_o} - \mathbf{P_n}\|} \mathbf{d_o}, \tag{8}$$

where $k_b$ was the rotational spring constant, $\theta_c$ the rest angle at $m_n$ with respect to cell $c$, $\theta$ the current signed angle made by the cell walls, and $\mathbf{d_o}$ was the direction of the force on $m_o$. The direction $\mathbf{d_o}$ was taken to be the outward facing normal to the wall-segment spanning $\mathbf{P_o}$ and $\mathbf{P_n}$. In simulations, we assumed that cell walls have a straight rest state (i.e. $\theta_c = \pi$). The force on $m_r$ had a similar form

$$\mathbf{F_{b_c}^r} = k_b \frac{(\theta_c - \theta)}{\|\mathbf{P_r} - \mathbf{P_n}\|} \mathbf{d_r}, \tag{9}$$

but the direction of the force was $\mathbf{d_r}$, the outward facing normal to the wall-segment spanning $\mathbf{P_r}$ and $\mathbf{P_n}$. To conserve momentum, the force on $m_n$ was

$$\mathbf{F_{b_c}^n} = -(\mathbf{F_{b_c}^o} + \mathbf{F_{b_c}^r}) \tag{10}$$

The force $\mathbf{F_b}$ for a mass $m_i$ was calculated by visiting all bending springs where $m_i$ appears as $m_o$, $m_r$ or $m_n$ and summing the corresponding force (as given by **Equations 8-10**).

Finally, let us consider the value of the force $\mathbf{F_{micro}}$, produced by the stretching of the microtubule based growth restrictions connected to mass $m_i$. This force was the sum of the forces $\mathbf{F_{micro_j}}$ resulting from the stretching of each growth-restricting spring connecting a mass $m_j$ to $m_i$, and had the same form as **Equation 7**:

$$\mathbf{F_{micro_j}} = -k_m \frac{l_{ij} - r_{ij}}{r_{ij}} \cdot \frac{\mathbf{P_i} - \mathbf{P_j}}{l_{ij}}, \tag{11}$$

but used a different tensional elastic modulus $k_m$. Similar to **Equation 7**, $r_{ij}$ was the rest length of the spring connecting $m_i$ and $m_j$, and $l_{ij} = \|\mathbf{P_i} - \mathbf{P_j}\|$ was the actual length of the spring. Compressive forces (i.e. $\|\mathbf{P_i} - \mathbf{P_j}\| < r_{ij}$) were ignored, to approximate the properties of cells (i.e. consistent with the high tensile strength of cellulose; **Cosgrove, 2005**). Additionally, the springs were assumed to be inactive when $r_{ij}$ was outside an active range related to the target LEC (i.e. the spring was only active when $min_{micro} < r_{ij} < max_{micro}$).

## Parameters

As the model is 2D, simulations only offer a qualitative representation of reality. We report the parameters used in simulations in **Appendix 1—table 1** to ensure reproducability and demonstrate the reasonableness of chosen parameter values. Reasonableness of chosen parameters is supported by the parameter space exploration reported in the main text (**Figure 3—figure supplements 1** and **2**), which illustrates that the model produces plausible cell shapes for a broad range of parameter values. We provide further discussion of simulation parameters below.

First, we note that model results do not depend on the absolute values of stiffness parameters ($k_s$, $k_b$ and $k_m$), but rather their relative values. This follows from the fact that mechanical equilibrium (i.e. when **Equations 4-5** are 0) depends on the relative value of the forces acting on masses. For the chosen stiffness values we obtain the results reported in the main text for reasonable anisotropy values (i.e. growth in the x and y directions always differ by less than an order of magnitude).

Due to the models mass-spring formulation, the precise connection to a continuum is unclear, and varying the edge subdivision threshold $Th_{sub}$ influences simulation results. However, similar results can be produced when the density of masses is changed by coordinated changes in parameters (varying both $k_m$ and $k_b$ in tandem). If the threshold $Th_{sub}$ is too large then cell shapes become unrealistic, due to sparse sampling of the cell contour. Consequently, we have chosen such values that further refinement does not give qualitatively different results.

