## [Decision Letter]

Thank you for submitting your article "Why plants make puzzle-shaped cells" for consideration by *eLife*. Your article has been reviewed by three peer reviewers, and the evaluation has been overseen by a Reviewing Editor and Christian Hardtke as the Senior Editor.

The reviewers have discussed the reviews with one another and the Reviewing Editor has drafted this decision to help you prepare a revised submission.

Summary:

Epidermal cells on many plant leaves have complex, interdigitating shapes. The mechanism by which these complex yet robust shapes are formed, and their functional significance has been the subject of much investigation, with various interpretations proposed, all with their own merits and problems. Here they propose that the need to contain or distribute mechanical stress in cells to prevent potential catastrophic breakdown has been a primary driver in the generation of this cellular pattern, and they provide a relatively simple mechanical stress-based model by which such catastrophic stress could be automatically avoided. A major challenge in this field is to provide conclusive experimental proof, and the paper does struggle on this point. However, the large amount of data/observations provided are consistent with their model, the simplicity and robustness of the model itself (compared to alternatives which invoke local cell/cell signaling, in which the nature of the mobile signal is very unclear/contentious) makes a strong case that their model captures an important element of the patterning system, both in terms of how lobed cells are generated and the underlying reason. The quality of the Supplemental videos really brings this work to life and make their key points in a very convincing manner.

Essential revisions:

1) Incorporate discussion of "Mechanochemical polarization of contiguous cell walls shapes plant pavement cells", by Majda et al., 2017 which provides some idea about how altered cytoskeletal/cellulose patterns might feed into differential ROP2/cell wall structure, allowing lobing to occur i.e., it addresses some of the mechanistic points raised below. There may be questions about cause and effect between the two papers, but this is a discussion point.

2) Modeling results are described in a highly non-quantitative or comparative manner making it impossible to judge the robustness of outcomes against parameter variations, or which parameters are most critical. In subsection “A mechanistic model of puzzle shape emergence” it says that the main parameters are cell wall stiffness, angle at which additional connections can be made and the convexity criteria for attaching to the opposing wall and that's it. The authors state that growth needs to be isotropic for puzzle-shapes to emerge. However, it is nowhere made clear how stringent this demand is, i.e. is an anisotropy ratio of 1:1 sufficient to prevent puzzle-shapes or does it need to be 10? More quantitative data and comparisons are called for, and not phrases like "somewhat isotropically" (subsection “Isotropic tissue growth is correlated with puzzle-shaped cell formation”). At the very least the authors should:

- Give a table listing all parameters and their default values.

- Indicate for results whether these parameter values or others have been used.

- Vary for all (major) parameters the used values over a range.

- Determine how modeling outcomes depend on these parameter values.

- Analyse which parameters are hence most critical.

- Argue why the used parameter values are valid.

It might be useful to present accessible simulation runs, like those shown in their movies, with a slider for their key parameters. This would allow users to readily validate (or not) how robust their models are.

3) A key part of the model is how opposite sides of a cell (with appropriate curvature) "know" what is happening. The role of ROP2/ROP6 is reasonable, but there are still obvious holes in our molecular/structural understanding of this process. Authors might want to highlight that this is an important mechanistic "unknown" in their model. They propose that additional springs are removed if they are connecting to concave sides and added if they are connecting to convex sides. Yes, ROP2 and ROP6 are mutually exclusive, and ROP2 links to actin and expansion and ROP6 to microtubules and constriction. However, this indicates that ROP type influences local curvature, but not the other way around, i.e. that curvature determines ROP species. It will eventually boil down to this as an end result, but how is it justified to use it as a causal agent unless there is a causal link between curvature and the type of ROP species accumulating?

4) In the analysis of the spike1 mutant, the epidermal cell area seems much larger than in WT – is this true? Even if WT puzzle cells attained this size they would not be expected to show the cell separation phenotype observed in spike1. Is this true? What is the modeled stress distribution in the spike1 epidermal cells, particularly around the circumference? Could it in anyway account for the cell separation phenotype? The link between spike1 cell shape and cell separation was unclear. Can the authors clarify? In subsection “A strategy for when lobes cannot be formed” the authors imply that they mimicked the spike1 phenotype by removing ROP2 only from their model. I think SPIKE1 is meant to modulate both ROP2 and ROP6, which are proposed to act in mutual inhibition. In the model, is loss of ROP2 alone functionally equivalent to the loss of both ROP2 and ROP6? Needs clarification.

5) Another key proposition of their model is that mechanical stress is focused on particular regions of the cell circumference. Is there any evidence of localized change in cell wall composition/architecture that might allow these regions to withstand/cope with these stress points? As the authors mention (subsection “Cell shape and size across species”), one way around the potential issue of excessive mechanical stress in the outer paradermal cell wall of the epidermis is to have a thicker/stronger cell wall, and, indeed, this wall is generally significantly thicker than other cell walls. In their model, is it possible to explore how much relatively thicker this cell wall would have to be to contain the relative increase in stress predicted to occur in this wall, i.e., to prevent bursting? Linked to this point, the authors state (Introduction) that the puzzle-shape cell shape benefits the plant by "lowering the amount of cellulose necessary to keep the integrity of the cell wall". Do the authors really provide any evidence for this conjecture? They don't need to invoke this reasoning. If the models hold true, then preventing cell bursting would appear to be the key feature (unless a very minor change in paradermal cell wall thickness would solve the problem). Also linked to this, their model starts with the assumption of the wall being homogenous (Introduction). This is a reasonable place to start but, obviously, is a major simplification (cell walls are not homogenous) and there are ample possibilities for local anisotropy/structure within a cell wall. This should be made explicit. Subsection “A mechanistic model of puzzle shape emergence” and following: New springs are envisioned as cellulose, yet pure cellulose is definitely not elastic! Maybe better to say that the preferred orientation of cellulose µFs is set by this parameter? Authors should avoid suggesting that cellulose can act as an an elastic spring. At some places they say that the additional across cell springs resist elongation, whereas also in subsection “A mechanistic model of puzzle shape emergence” it seems to say they do not grow at all. So, what exactly is done with the length of these additional springs, e.g. in the updating where new ones can be added and old ones removed based on curvature, are lengths adjusted or not?

They propose that microtubules (aka additional springs) only form across the cell and not from one indentation to the next on the same side. Why is this a reasonable assumption? Because microtubules and actin do not cross? Because this would imply odd bending for microtubules? Subsection “Experimental evidence that stress needs to be managed”: Something that depolymerizes cortical microtubules interferes with cellulose deposition and thus strongly impacts wall stiffness and is not some minor thing that can be excluded.

6) In a number of the figures (e.g., Figure 3) the authors refer to maximal stress. Do they mean maximal stress values in the outer paradermal cell wall or are these actually mean stress values calculated for all surfaces of the cell? The key point from the images is the localization of the max stress value and how this is dissipated/decreased by the change of cell shape. It needs to be clear from the figures and associated legends which "stress" values the graphs are referring to.

7) Subsection “Cell shape predicts mechanical stress magnitude” and in the Discussion section – note that in some leaves (grasses) growth is clearly anisotropic. Consistent with their hypothesis, longitudinal division maintains a similar anisotropy in the un-lobed long epidermal cells. However, in some regions of the grass epidermis (often adjacent to veins) cells do occur which have some lobing. This may reflect special topography or growth vector in this region. Main point is that not all leaves grow isotropically.

---

## [Author Response]

Summary:Epidermal cells on many plant leaves have complex, interdigitating shapes. The mechanism by which these complex yet robust shapes are formed, and their functional significance has been the subject of much investigation, with various interpretations proposed, all with their own merits and problems. Here they propose that the need to contain or distribute mechanical stress in cells to prevent potential catastrophic breakdown has been a primary driver in the generation of this cellular pattern, and they provide a relatively simple mechanical stress-based model by which such catastrophic stress could be automatically avoided. A major challenge in this field is to provide conclusive experimental proof, and the paper does struggle on this point. However, the large amount of data/observations provided are consistent with their model, the simplicity and robustness of the model itself (compared to alternatives which invoke local cell/cell signaling, in which the nature of the mobile signal is very unclear/contentious) makes a strong case that their model captures an important element of the patterning system, both in terms of how lobed cells are generated and the underlying reason. The quality of the Supplemental movies really brings this work to life and make their key points in a very convincing manner.Thank you for the review. We believe we have been able to address all of the reviewers’ comments. Our response to each comment is below in blue text. We have indicated the line numbers where changes were made in the manuscript corresponding to each point. We have also added a few additional references to new work that has become available since our initial submission.Essential revisions:1) Incorporate discussion of "Mechanochemical polarization of contiguous cell walls shapes plant pavement cells", by Majda et al., 2017 provides some idea about how altered cytoskeletal/cellulose patterns might feed into differential ROP2/cell wall structure, allowing lobing to occur i.e., it addresses some of the mechanistic points raised below. There may be questions about cause and effect between the two papers, but this is a discussion point.

We have now added discussion of, and references to, Majda et al., (2017) throughout the manuscript. We provide a detailed discussion of the relation between our model and that of Majda et al., (2017) in the Discussion section.

We have also taken this opportunity to introduce references to and discussion of other recent articles the Introduction and the Discussion section.

2) Modeling results are described in a highly non-quantitative or comparative manner making it impossible to judge the robustness of outcomes against parameter variations, or which parameters are most critical. In subsection “A mechanistic model of puzzle shape emergence” it says that the main parameters are cell wall stiffness, angle at which additional connections can be made and the convexity criteria for attaching to the opposing wall and that's it. The authors state that growth needs to be isotropic for puzzle-shapes to emerge. However, it is nowhere made clear how stringent this demand is, i.e. is an anisotropy ratio of 1:1 sufficient to prevent puzzle-shapes or does it need to be 10? More quantitative data and comparisons are called for, and not phrases like "somewhat isotropically" (subsection “Isotropic tissue growth is correlated with puzzle-shaped cell formation”). At the very least the authors should:

To clarify the importance of model parameters to simulation results and highlight the robustness of model outcomes we have performed a systematic exploration of key parameters and made parameter values used in simulations more apparent.

At the very least the authors should:- Give a table listing all parameters and their default values.

A parameter table for all parameters is provided along with a description of the model implementation in Supplementary File 1, and a reference to it has been added to subsection “A mechanistic model of puzzle shape emergence”.

- Indicate for results whether these parameter values or others have been used.

In addition to specifying values in the parameter table we now also explicitly indicate which parameters have been changed between simulations. Please see the revisions introduced on lines subsection “A mechanistic model of puzzle shape emergence”, the caption of Figure 3—figure supplement 1), and subsection “A strategy for when lobes cannot be formed”.

- Vary for all (major) parameters the used values over a range.- Determine how modeling outcomes depend on these parameter values.-Aanalyse which parameters are hence most critical.

We have now performed a parameter space exploration for critical parameters.

The outcome is summarized in a new figure (Figure 3—figure supplement 1), which contains an array of model outcomes where the most important parameters are varied. In this context “important” is defined as those parameters that have the greatest impact on cell shape in the simulation.

In addition to the text for the caption of Figure 3—supplement 1 and additional videos (Video 5–Video 10), discussion of the analysis can be found in subsection “A mechanistic model of puzzle shape emergence”.

- Argue why the used parameter values are valid.

As the model is 2D, the simulation (mechanically) only offers a qualitative representation of reality. Having said that, parameters, such as the cell wall spring stiffness could be matched to an estimate for the cell wall Young’s modulus, however data are scarce in this area. This is not of great concern, since the model does not depend on the absolute value of this parameter, but rather the ratio of the wall stiffness to the stiffness of the cross springs that represent the cellulose microfibril bundles, thus any value for Young’s modulus could be fit to the model. However, the ratio can be tested for reasonableness in the sense that extreme values for anisotropy would be a red flag. As can be seen from the parameter table, these two parameters differ by less than one order of magnitude, and thus extreme values of anisotropy are not required.

We have also added discussion of this in the supplement where the parameters are covered in detail (Supplementary File 1, section 2.6).

It might be useful to present accessible simulation runs, like those shown in their movies, with a slider for their key parameters. This would allow users to readily validate (or not) how robust their models are.

This idea is very nice, and although it would be difficult to provide a running version for multiple platforms, we think we have a found a simple alternative. We have created several Videos (see Video 5–Video 10) showing the effect of varying the most important parameters. In these movies, the final time point of the simulation is shown while varying a single parameter. Although this is not a live simulation, it allows a very quick and easy way to explore the model parameter space on any platform.

We also note that we will provide the simulation source programs upon request. We will help people (within reason) to get it working, which requires Linux, the g++ tool chain, libraries, and so on. This has been noted in the revised supplementary information (Supplementary File, section 2.1).

3) A key part of the model is how opposite sides of a cell (with appropriate curvature) "know" what is happening. The role of ROP2/ROP6 is reasonable, but there are still obvious holes in our molecular/structural understanding of this process. Authors might want to highlight that this is an important mechanistic "unknown" in their model.

As lobes and indentations start to form, this creates a feedback via the shape on the placement of the cross springs (representing the cellulose bundles) in the model. An (active) restriction in one cell creates a (passive) protrusion in the neighbor. The ROPs, in combination with microtubule dynamics, are likely involved in this shape enhanced feedback in the indentations, and also indirectly in the lobes as they repress each other. Although we do not rule out a mobile signal crossing the wall to synchronize lobes with indentations, our model demonstrates that puzzle-shaped epidermal cells can emerge without such a signal.

As the reviewer suggests, we now explicitly state that the molecular mechanism to sense stress (or strain) and its relation to the ROPs is unknown in the Discussion section.

They propose that additional springs are removed if they are connecting to concave sides and added if they are connecting to convex sides. Yes, ROP2 and ROP6 are mutually exclusive, and ROP2 links to actin and expansion and ROP6 to microtubules and constriction. However, this indicates that ROP type influences local curvature, but not the other way around, i.e. that curvature determines ROP species. It will eventually boil down to this as an end result, but how is it justified to use it as a causal agent unless there is a causal link between curvature and the type of ROP species accumulating?

The key idea here is that if stress triggers a feedback process where indentations become more indented, and this involves ROP6, then ROP2 gets repressed in the lobes because ROP6 is in the indentations nearby. From this perspective, lobes are not actively "created", but rather emerge as a result of being between two indentations. It also means that one is not really downstream of the other. If ROP6 is part of the stress sensing mechanism, the two processes act together. The Abley et al., (2013) model relies on the auto-activation of membrane-bound ROP6, and the stress feedback mechanism may be a part of that.

Feedbacks between geometry and molecular activity to produce regular outgrowths have been observed in other systems. An example from animals is found in epithelial folding underlying villi formation in developing guts (Shyer et al., (2015), where growth induced folding changes morphogen gradients based on local geometry. Increased morphogen concentration at incipient protrusions (villi) then locally modifies growth to further focus the morphogen gradient and accentuate outgrowth.

We have changed the text to make these dynamics more explicit and mentioned the possibility of geometry sensing as in the gut (subsection “A mechanistic model of puzzle shape emergence” and the Discussion section).

4) In the analysis of the spike1 mutant, the epidermal cell area seems much larger than in WT – is this true?

The average cell size in *spike1* is larger than in wt only for the final time point of our measurements (6 DAG).

To clarify we have added a plot (Figure 5—figure supplement 3) showing mean average cell areas for wild type and *spike1* cotyledons and discussed it in the text in subsection “A strategy for when lobes cannot be formed”.

Even if WT puzzle cells attained this size they would not be expected to show the cell separation phenotype observed in spike1. Is this true?

This is correct, however *spike1* does not form lobes. Consequently, to obtain a similar LEC (as we show to be the case in Figure 5 and Figure 5—figure supplement 1) it forms worm-shaped cells instead. Models indicate that these cell shapes can be produced by removing the convexity condition and also increasing stiffness of connections per cell wall. This would increase stresses and may contribute to a cell-separation phenotype.

We observed that the cell adhesion defects appear very early in the development of cotyledon epidermis, before the cells become too large. We therefore think that the holes in the tissue may not be primarily related to increased stresses from cell shape, for example through the impact of actin-driven pectin delivery to cell walls, although increased stress may aggravate the situation. This is discussed in subsection “A strategy for when lobes cannot be formed”.

What is the modeled stress distribution in the spike1 epidermal cells, particularly around the circumference?

As in the other simulations, it scales with the LEC, which *spike1* manages to keep low in spite of being unable to make lobes. This is consistent with our assertion that long-thin “wormy” cells also provide a strategy to minimize LEC and cellular stresses.

We have added a figure showing stress distribution in *spike1* epidermis (Figure 5—figure supplement 2), with the simulation template coming from real cell shapes. We have also mentioned that stress values in *spike1* cells are comparable to those of wild type in subsection “A strategy for when lobes cannot be formed”.

Could it in anyway account for the cell separation phenotype? The link between spike1 cell shape and cell separation was unclear. Can the authors clarify? On subsection “A strategy for when lobes cannot be formed” the authors imply that they mimicked the spike1 phenotype by removing ROP2 only from their model. I think SPIKE1 is meant to modulate both ROP2 and ROP6, which are proposed to act in mutual inhibition. In the model, is loss of ROP2 alone functionally equivalent to the loss of both ROP2 and ROP6? Needs clarification.

As mentioned in the previous point, the phenotype appears very early, before large stresses could accumulate. While we agree it is appealing to think that the inability to make puzzle shaped cells drive this phenotype, at present the data do not provide convincing evidence for this scenario.

In light of this, we have changed the text to make it clear that we have no evidence that the cell shape is the primary driver of the cell-cell adhesion phenotype (subsection “A strategy for when lobes cannot be formed”).

With regards to models, as noted above, reproducing the *spike1* phenotype involves several parameter changes – consistent with broad defects in ROP-mediated cytoskeletal rearrangement, beyond just the loss of ROP2 localization. We now make this connection more explicitly (subsection “A strategy for when lobes cannot be formed”).

5) Another key proposition of their model is that mechanical stress is focused on particular regions of the cell circumference. Is there any evidence of localized change in cell wall composition/architecture that might allow these regions to withstand/cope with these stress points?

The AFM work of Sampathkumar et al., (2014) shows that cellulose bundles do accumulate in the high-stress indentations. This would be expected to counteract the higher stress in these regions. Sampathkumar et al., suggest that microtubles themselves can respond to stress, which gives a natural mechanism to reinforce high stress areas via their control of the direction of the cellulose synthase.

We have changed the text to highlight these points (subsection “A mechanistic model of puzzle shape emergence” and the Discussion section) in the context of discussions of Majda et al., (2017) which also provides evidence for localized changes in cellwall composition.

As the authors mention (subsection “Cell shape and size across species”), one way around the potential issue of excessive mechanical stress in the outer paradermal cell wall of the epidermis is to have a thicker/stronger cell wall, and, indeed, this wall is generally significantly thicker than other cell walls. In their model, is it possible to explore how much relatively thicker this cell wall would have to be to contain the relative increase in stress predicted to occur in this wall, i.e., to prevent bursting?

Estimating wall strength, or when the wall will fail *in planta*, is difficult and currently is poorly understood. The oryzalin experiments suggest that cells with an order of magnitude larger radius will explode, but it is questionable if these represent normal epidermal cells.

As mentioned above, one way to make the wall stronger at high stress points is to embed more cellulose in it. This would not have to necessarily change the thickness of the wall.

From a theoretical perspective, in a sphere, the force acting on a cross-section increases as the square of the radius, whereas the material holding back that force only increases linearly. Consequently, as the cells get larger, the stress increases linearly (for uniform wall thickness). Thus, the cell wall would have to increase in thickness linearly with cell (or LEC) diameter. As cells may increase over 100-fold in size during development this suggests that resisting stress exclusively by increasing wall thickness would require a substantial increase in wall thickness.

Linked to this point, the authors state (Introduction) that the puzzle-shape cell shape benefits the plant by "lowering the amount of cellulose necessary to keep the integrity of the cell wall". Do the authors really provide any evidence for this conjecture?

It follows that reducing stress would reduce the amount of cell wall material (including cellulose) required to counteract that stress.

We have rephrased the Discussion section accordingly.

They don't need to invoke this reasoning. If the models hold true, then preventing cell bursting would appear to be the key feature (unless a very minor change in paradermal cell wall thickness would solve the problem). Also linked to this, their model starts with the assumption of the wall being homogenous (Introduction). This is a reasonable place to start but, obviously, is a major simplification (cell walls are not homogenous) and there are ample possibilities for local anisotropy/structure within a cell wall. This should be made explicit.

We have mentioned this in the Discussion section and subsection “Cell shape predicts mechanical stress magnitude”.

Subsection “A mechanistic model of puzzle shape emergence” and following: New springs are envisioned as cellulose, yet pure cellulose is definitely not elastic! Maybe better to say that the preferred orientation of cellulose µFs is set by this parameter? Authors should avoid suggesting that cellulose can act as an elastic spring.

We have changed the text to clarify that the springs represent the presence of oriented cellulose in the cell wall matrix, and possibly other components that lead to wall anisotropy and growth restriction, rather than the individual cellulose fibers or bundles themselves (subsection “A mechanistic model of puzzle shape emergence”).

At some places they say that the additional across cell springs resist elongation, whereas also in subsection “A mechanistic model of puzzle shape emergence” it seems to say they do not grow at all. So, what exactly is done with the length of these additional springs? e.g. in the updating where new ones can be added, and old ones removed based on curvature, are lengths adjusted or not?

The additional springs resist elastic elongation, but their reference configuration does not change. In this context growth refers to changes in the reference configuration. Reversible deformations are not considered as growth. Lengths are determined when the additional springs are placed and not modified. We have rephrased to clarify this point in subsection “A mechanistic model of puzzle shape emergence”.

They propose that microtubules (aka additional springs) only form across the cell and not from one indentation to the next on the same side. Why is this a reasonable assumption? Because microtubules and actin do not cross? Because this would imply odd bending for microtubules?

These rules result in patterns of cellulose deposition that appear to follow the patterns of stress, as previously reported in Sampathkumar et al., (2014). Notably, their results indicate that stresses in the periclinal wall tend to orient away from, not along, the anticlinal wall, as we assume in models.

We have added a comment on this point in subsection “A mechanistic model of puzzle shape emergence”.

Subsection “Experimental evidence that stress needs to be managed”: Something that depolymerizes cortical microtubules interferes with cellulose deposition and thus strongly impacts wall stiffness and is not some minor thing that can be excluded.

Oryzalin depolymerizes microtubules, and the assumption is that cellulose deposition still proceeds (Paredez et al., 2006), however there is no guarantee that it proceeds as the same rate as before. In Arabidopsis hypocotyl it has been observed that oryzalin does change the direction of cellulose deposition (becomes more isotropic) but does not affect deposition rates (Chan et al., 2010). In elongated cells microtubule depolymerization causes loss of growth anisotropy, but since meristem growth is primarily isotropic, we assume its biggest effect in this case is to stop cell division (and thus enlarge cells). Even if the amount of cellulose is compromised, the observation that larger cells (at the flank of the meristem) are more likely to burst than smaller cells (in the middle of the meristem) still holds.

We have clarified the discussion on the effects of oryzalin and mentioned that it is possible that is could also reduce cellulose deposition in subsection “Experimental evidence that stress needs to be managed”.6) In a number of the figures (e.g., Figure 3) the authors refer to maximal stress. Do they mean maximal stress values in the outer paradermal cell wall or are these actually mean stress values calculated for all surfaces of the cell? The key point from the images is the localization of the max stress value and how this is dissipated/decreased by the change of cell shape. It needs to be clear from the figures and associated legends which "stress" values the graphs are referring to.

We have made this explicit in the caption (subsection “A mechanistic model of puzzle shape emergence”), and we have also more precisely defined the measure of stress, since it is a scalar value computed from a tensor (subsections “Cell shape predicts mechanical stress magnitude”, “A mechanistic model of puzzle shape emergence” and Figure 5—figure supplement 3).

7) Subsection “Cell shape predicts mechanical stress magnitude” and in the Discussion section – note that in some leaves (grasses) growth is clearly anisotropic. Consistent with their hypothesis, longitudinal division maintains a similar anisotropy in the un-lobed long epidermal cells. However, in some regions of the grass epidermis (often adjacent to veins) cells do occur which have some lobing. This may reflect special topography or growth vector in this region. Main point is that not all leaves grow isotropically.

We agree that this is likely due to the particular growth field that cells experience in these areas. During the parameter analysis we found conditions that create shapes similar to grasses (see Figure 3—figure supplement 1).

We have added a comment on this in the text (subsection “Cell shape predicts mechanical stress magnitude” and the Discussion section).